# A novel class of inhibitors that disrupts the stability of integrin heterodimers identified by CRISPR-tiling-instructed genetic screens

Nicole M. Mattson [1], Anthony K. N. Chan [1,2], Kazuya Miyashita[1], Elizaveta Mukhaleva[3], Wen-Han Chang[1], Lu Yang[1,2], Ning Ma [3], Yingyu Wang[3], Sheela Pangeni Pokharel[1,2], Mingli Li[1], Qiao Liu[1], Xiaobao Xu[1], Renee Chen[1], Priyanka Singh[1], Leisi Zhang[1], Zeinab Elsayed[1], Bryan Chen[1], Denise Keen[4], Patrick Pirrotte[5,6], Steven. T. Rosen[4], Jianjun Chen [1,4], Mark A. LaBarge[4,7], John E. Shively[4,8], Nagarajan Vaidehi [3,4], Russell C. Rockne [3,4], Mingye Feng[4,9] & Chun-Wei Chen [1,2,4] ✉

The plasma membrane is enriched for receptors and signaling proteins that are accessible from the extracellular space for pharmacological intervention. Here we conducted a series of CRISPR screens using human cell surface proteome and integrin family libraries in multiple cancer models. Our results identified ITGAV (integrin αV) and its heterodimer partner ITGB5 (integrin β5) as the essential integrin α/β pair for cancer cell expansion. High-density CRISPR gene tiling further pinpointed the integral pocket within the β-propeller domain of ITGAV for integrin αVβ5 dimerization. Combined with in silico compound docking, we developed a CRISPR-Tiling-Instructed Computer-Aided (CRISPR-TICA) pipeline for drug discovery and identified Cpd_AV2 as a lead inhibitor targeting the β-propeller central pocket of ITGAV. Cpd_AV2 treatment led to rapid uncoupling of integrin αVβ5 and cellular apoptosis, providing a unique class of therapeutic action that eliminates the integrin signaling via heterodimer dissociation. We also foresee the CRISPR-TICA approach to be an accessible method for future drug discovery studies.

The plasma membrane is a semipermeable barrier that encloses intracellular components from the extracellular environment. In addition to the phospholipid bilayer, proteins are estimated to constitute as much as 50% of plasma membrane biomass. The cell surface proteome can be divided into integral membrane proteins (transmembrane proteins) and peripheral membrane proteins (membrane-anchored and other cell-surface-associated proteins), which are highly enriched for proteins involved in cellular adhesion, migration, communication, ligand binding, signal transduction, nutrition/ion transport and immunity[1,2]. An estimated 543 to 1,100 different proteins are present on the cancer

[1]Department of Systems Biology, Beckman Research Institute, City of Hope, Duarte, CA, USA. [2]Division of Epigenetic and Transcriptional Engineering, Beckman Research Institute, City of Hope, Duarte, CA, USA. [3]Department of Computational and Quantitative Medicine, Beckman Research Institute, City of Hope, Duarte, CA, USA. [4]City of Hope Comprehensive Cancer Center, Duarte, CA, USA. [5]Integrated Mass Spectrometry Shared Resource, City of Hope Comprehensive Cancer Center, Duarte, CA, USA. [6]Cancer and Cell Biology Division, Translational Genomics Research Institute, Phoenix, AZ, USA. [7]Department of Population Sciences, Beckman Research Institute, City of Hope, Duarte, CA, USA. [8]Department of Immunology and Theranostics, Beckman Research Institute, City of Hope, Duarte, CA, USA. [9]Department of Immuno-Oncology, Beckman Research Institute, City of Hope, Duarte, CA, USA. ✉e-mail: cweichen@coh.org

cell surface[3], many of which have functions that could influence disease progression and therapeutic response. Owing to the extracellular accessibility and the substantial biological functions of plasma membrane proteins, the cell surface proteome represents a valuable pool of targets for pharmacological intervention[4,5].

Among these potential targets are integrins, a family of cell surface transmembrane receptors that have important roles in cell-to-cell and cell-to-extracellular-matrix (ECM) interactions[6]. Activation of integrins also controls the morphology, polarity and migration of cells by engaging the cells to the extracellular environments and by rearranging intracellular cytoskeleton components such as actin filaments (that is, outside-in signaling)[7]. Integrins can also be activated through their carboxyl terminus intracellular tails to engage extracellular ligands (that is, inside-out signaling)[8]. Thus far, 24 distinct integrins have been documented, in four subfamilies (the RGD receptors, collagen receptors, laminin receptors and leukocyte-specific receptors), each of which is a cell surface heterodimer comprising of one of the 18 α subunits and one of the eight β subunits in the human genome[6]. These diverse integrin α/β pairs govern tissue morphogenesis, homeostasis, angiogenesis, thrombosis and inflammatory response[6,9]. In addition, integrins including α4β1, α5β1, αVβ3 and αVβ5 have been implicated in the carcinogenesis and metastasis of various tumor types[10]. Therefore, pharmacological targeting of specific integrin α/β pairs has become an attractive field for therapeutic development[9,11].

CRISPR–Cas9 (clustered regularly interspaced short palindromic repeats and CRISPR-associated endonucleases)[12–14] gene suppression screens are powerful genetic approaches for identifying effector genes in biological systems[15–17]. For example, the DepMap consortium (https://depmap.org/portal/; BROAD Institute) has performed genome-wide CRISPR library knockout screens in ~1,000 cell line models and enabled the discovery of a wide range of cancer-cell-dependent genes[18,19]. Furthermore, recent advances in high-density CRISPR gene tiling have revealed the utility of CRISPR technology in protein domain and subdomain characterization[20–22]. The saturation CRISPR mutagenesis screen thus provides a powerful platform for examining critical functional areas within the protein of interest and could instruct discovery of therapeutics.

In this study, we conducted a series of unbiased CRISPR screens, including a surface proteome library screen and an integrin-family-focused library screen. As a result, we identified a requirement for integrin αV (ITGAV) and integrin β5 (ITGB5) in tumor cell maintenance. We also exploited the power of high-density CRISPR gene-tiling screens[23–28] and developed a drug discovery tool named CRISPR-TICA (CRISPR-Tiling-Instructed Computer-Aided). This workflow allowed us to identify an integrin αVβ5 disruptor that targets the CRISPR-hypersensitive pocket within the β-propeller domain of ITGAV to prevent its interaction with ITGB5, providing a unique class of integrin inhibitor that acts through dissociation of the α/β heterodimers.

## Results

### Cell surface proteome CRISPR library screens
To identify critical cell surface proteins required for cancer cell expansion, we evaluated a mass-spectrometry-derived cell surface protein atlas (1,492 genes)[4] together with a membrane protein database (2,418 genes)[29] and the Human Protein Atlas – Plasma Membrane (2,254 genes; https://www.proteinatlas.org) and summarized 581 core cell surface proteins (Supplementary Table 1). Based on this, we developed a focused CRISPR library targeting the 581 genes encoding these cell surface proteins (involved in cellular adhesion, migration, communication, ligand binding, signal transduction, nutrition/ion transport, immunity and so on) with 2,905 single-guide RNAs (sgRNAs) (Fig. 1a; five sgRNAs per gene). We also spiked in a panel of 41 negative control sgRNAs (targeting nonhuman genes such as *Luc*, *LacZ*, *Ren* and *Rosa26* and scrambled sequences) and 27 positive control sgRNAs (targeting cancer-essential genes such as *MYC*, *PCNA* and *RPA3*) (Extended Data

Fig. 1a,b and Supplementary Table 2). We then delivered this library into five Cas9-expressing human cancer cell lines (MDA231, PANC1, U251, SW620 and H661) using lentiviral transduction and compared the change in frequency of each integrated sgRNA construct in these cells between day 0 and day 24 using high-throughput sequencing followed by the MAGeCK algorithm[30] (Fig. 1b and Supplementary Table 3). Combined analyses revealed that in addition to proteins commonly required for cell proliferation (positive controls; yellow dots), these CRISPR screens identified ITGAV (encoded by the *ITGAV* gene) as the top essential cell surface protein in multiple cells representative of cancer types (Fig. 1c and Supplementary Table 3; additional analyses of these surface proteome screens are shown in Extended Data Fig. 2).

To validate the library screen results, we transduced the Cas9+ MDA231 and PANC1 cells with sgRNAs targeting *ITGAV* (sgITGAV) to deplete the expression of endogenous ITGAV (Fig. 1d). Using a flow cytometric growth competition assay (Extended Data Fig. 3a), we found that cells transduced with sgITGAV were outcompeted by cells transduced with sgRNA targeting nonessential sequences (sgCtrl) (Fig. 1e), and further expression of an exogenous *ITGAV* complementary DNA (cDNA) rescued the cells from sgITGAV (Extended Data Fig. 4). CRISPR depletion of ITGAV in MDA231-Cas9+ cells also led to pronounced apoptosis (Fig. 1f) and arrested the cell cycle (Fig. 1g). Clinically, we found that ITGAV was overexpressed in multiple cancer types, including those tested in our cell surface proteome CRISPR screens (Supplementary Fig. 1). We also observed an association of high ITGAV expression with poor survival prognosis in patients with diverse cancer types (Fig. 1h; source: Gene Expression Profiling Interactive Analysis (GEPIA), including breast carcinoma, pancreatic adenocarcinoma, lung adenocarcinoma, hepatocellular carcinoma and glioma; total: ~3,700 patients)[31], highlighting the requirement for ITGAV in cancer progression.

### ITGAV mediates RAC1 signaling and F-actin assembly
Integrins are known to control cytoskeletal rearrangement via plasma-membrane-associated Rho family small GTPases, including RHOA, RAC1 and CDC42 (ref. 32). To elucidate the cytoskeletal signaling pathways affected by depletion of ITGAV, we analyzed the DepMap genome-wide CRISPR screen consortium database (https://depmap.org/portal/; BROAD Institute) and found higher correlations of CERES scores (CERES is a computational method to estimate gene-dependency levels from CRISPR–Cas9 essentiality screens)[18] between *ITGAV* and *RAC1* in the 769 tested cell models (Fig. 2a, purple, and Extended Data Fig. 5a; Pearson coefficient = 0.482; rank 6 of 17,709 genes) than between *ITGAV* and *CDC42* or *RHOA* (green). The code-pendency relationship between *ITGAV* and *RAC1* indicate that ITGAV primarily signals through RAC1 to mediate the intracellular response. We then performed RNA sequencing and gene set enrichment analysis (GSEA)[33] on MDA231-Cas9+ cells transduced with sgCtrl versus sgITGAV. According to this transcriptomic analysis, the 'RAC1_GTPase_Cycle' was among the most depleted gene sets following sgITGAV transduction (Fig. 2b and Supplementary Table 4), supporting the data shown in Fig. 2a. Similar to sgITGAV, depletion of RAC1 by CRISPR (Fig. 2c) suppressed proliferation and survival of MDA231-Cas9+ cells (Fig. 2d–f). To examine the impact of ITGAV on cytoskeletal dynamics, we stained actin filaments (F-actin) with Alexa Fluor 488-conjugated phalloidin in MDA231-Cas9+ cells. Similar to sgRAC1 (purple), sgITGAV transduction (red) resulted in altered cellular morphology, disassembled cytoskeleton and reduced cell size (Fig. 2g,h). Overall, our results indicate an essential role of ITGAV/RAC1 signaling in cancer cell maintenance.

### Integrin family CRISPR screens highlighted integrin αVβ5
Integrins are obligate heterodimeric cell surface receptors composed of one of the 18 α subunits and one of the eight β subunits in the human genome (Fig. 3a)[6]. To pinpoint the critical components within the integrin family that collaborate with ITGAV for cancer cell maintenance, we developed another CRISPR library targeting the gene coding regions of

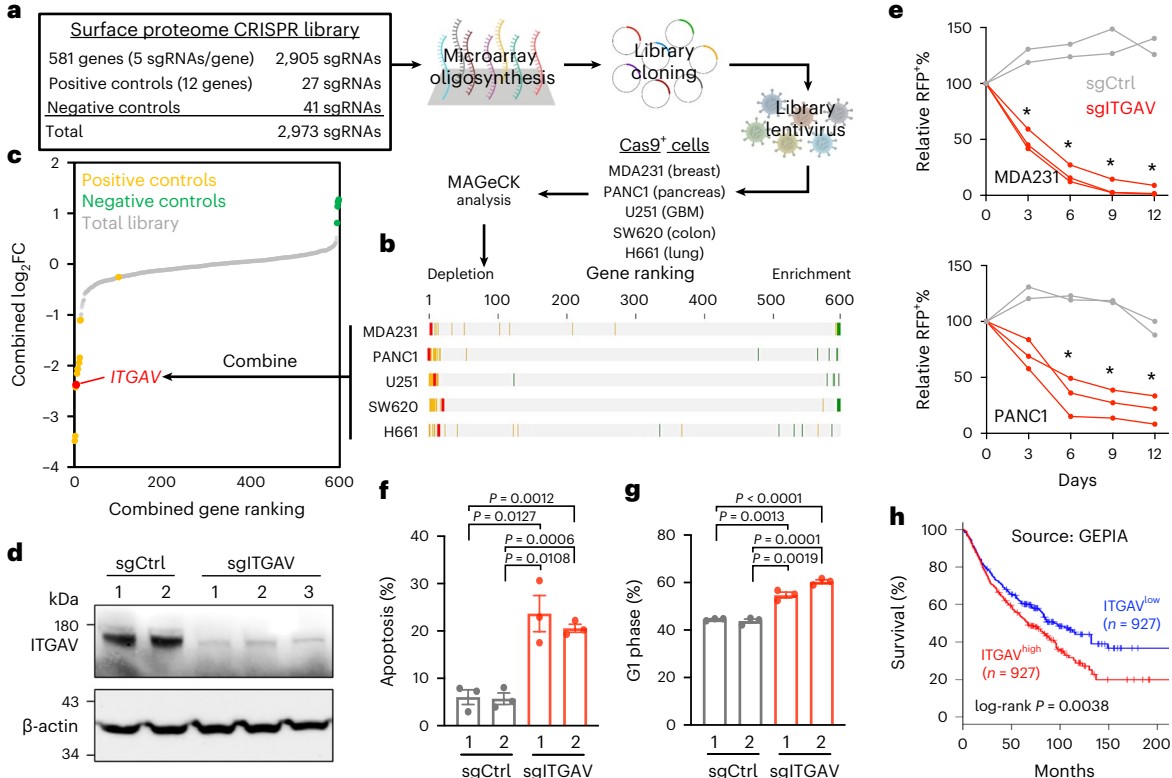

**Fig. 1 | Cell surface proteome CRISPR screens identify the essential role of ITGAV in cancer cells. a**, Schematic outline of cell surface proteome CRISPR screens (2,973 sgRNAs) in Cas9-expressing cancer cell models. **b,c**, Gene rankings for cell surface proteome CRISPR screens in five individual cell models (**b**) and the combined analysis (**c**) as calculated by the MAGeCK algorithm. The rankings of *ITGAV* (red), positive controls (yellow), negative controls (green) and the total library (gray) are indicated. **d**, Western blot of ITGAV and β-actin in MDA231-Cas9+ cells transduced with sgCtrl (*n* = 2 independent sgRNA sequences) and sgITGAV (*n* = 3 independent sgRNA sequences) for 3 days. **e**, Growth competition assay of MDA231-Cas9+ and PANC1-Cas9+ cells transduced with RFP-labeled sgCtrl (gray lines; two independent sgRNA sequences) and sgITGAV (red lines; three independent sgRNA sequences). Asterisk indicates that all three sgITGAV groups were significantly different (*P* < 0.01) from the two sgCtrl groups (*n* = 3 for each group). **f,g**, Cellular apoptosis detected by Annexin V+/DAPI− (**f**) and cell cycle monitored by EdU incorporation (**g**) in MDA231-Cas9+ cells transduced with sgCtrl and sgITGAV for 3 days (*n* = 3 for each group). **h**, Survival curves for cancer patients with high (top quartile; *n* = 927) versus low (bottom quartile; *n* = 927) ITGAV expression (data source: GEPIA). Data are presented as the mean ± s.e.m. *P* values were calculated by two-sided Student's *t*-test. FC, fold change; GBM, glioblastoma multiforme.

each of the 26 integrin subunits with 25 sgRNAs per gene (Extended Data Fig. 1c and Supplementary Table 2; total 650 sgRNAs plus control sgRNAs) for a CRISPR depletion screen in the Cas9+ MDA231 and PANC1 cells (Fig. 3b). These validation screens suggested that in addition to those targeting *ITGAV*, the sgRNAs targeting *ITGB5* were strongly depleted in both cancer cell models (Fig. 3c and Supplementary Fig. 2). We then annotated the CRISPR impact score (the median log10 fold change of the 25 sgRNAs for each subunit) to the integrin heterodimer network (a total of 24 different integrins, each with a unique α/β combination)[6] and identified integrin αVβ5 (an RGD receptor mediating cell-to-ECM interaction) as the top essential integrin pair in these cancer cells (Fig. 3d, red dotted circle). CRISPR depletion of ITGB5 (but not the other ITGAV heterodimer partners, including ITGB1/3/6/8) suppressed proliferation of MDA231 cells (Fig. 3e). Similar to the results for sgITGAV, depletion of ITGB5 (Fig. 3f) impaired the survival and cell cycle of MDA231 cells (Fig. 3g,h), phenocopying the effect of sgITGAV (Fig. 1f,g). Further analysis of gene codependency in the DepMap genome-wide CRISPR screen consortium database showed that the highest correlation of CERES scores in the 769 tested cell models was that between *ITGAV* and *ITGB5* (Fig. 3i, blue; Extended Data Fig. 5b; Pearson coefficient = 0.686; rank 1 of 17,709 genes). By contrast, the correlations of CERES scores between *ITGAV* and other partner β subunit coding genes (*ITGB1/3/6/8*) were much weaker (Fig. 3i, yellow; Extended Data Fig. 5b), suggesting a selective requirement for the integrin αVβ5 heterodimer in cancer cell expansion.

## CRISPR gene tiling pinpointed the β-propeller domain of ITGAV

To define the regions of ITGAV required for cancer cell survival, we employed a high-density gene tiling scan that pinpoints the functional regions within a protein by CRISPR-induced mutagenesis[23–28]. For this, we constructed a CRISPR library of 348 sgRNAs that target every position with an NGG protospacer adjacent motif within the *ITGAV* coding exons (Fig. 4a, Extended Data Fig. 1d and Supplementary Table 2). We then utilized lentiviral transduction to deliver this library into the Cas9-expressing MDA231 cells and detected the frequencies of each sgRNA sequence before versus after 24 days of culture by NextSeq550 sequencing (Supplementary Table 5). After smoothen modeling of the local sgRNAs[34], our CRISPR-tiling scan identified seven regions with critical roles (Fig. 4b, numbers 1–7) within the β propeller domain of ITGAV (dashed box; F31–A467). We then mapped the normalized CRISPR score (NCS) on a three-dimensional (3D) structure of ITGAV (Fig. 4c; AlphaFold ID: P06756)[35]. We found that the CRISPR-sensitive peptide regions within the β-propeller domain represented the tips of the seven blade-shaped structures facing toward the aromatic residue-enriched central cavity, designated 'hypersensitivity-illustrated pocket' or 'HIP' (Fig. 4d; including F51, W123, F189, Y254, F308, S372 and Y436), which is in direct contact with a basic amino acid (lysine or arginine) in the loop motif of the βA domain within the β integrin subunits[36–38].

To investigate the role of the β-propeller HIP in the ITGAV–ITGB5 interaction in living cells, we employed a bioluminescence resonance

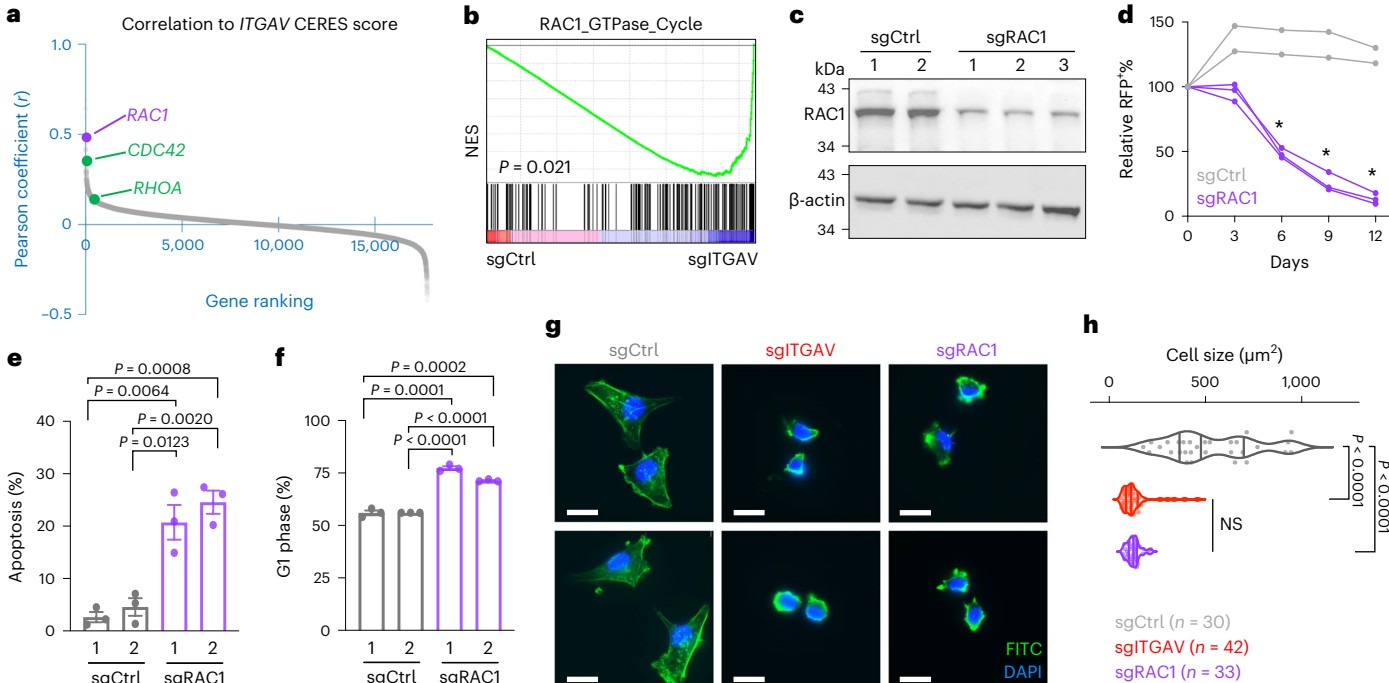

**Fig. 2 | ITGAV supports cancer cell expansion through small GTPase RAC1.**
**a**, Gene ranking based on the Pearson coefficient ($r$) of CERES scores between
*ITGAV* and *RAC1* (purple) compared with *CDC42* and *RHOA* (green) in the 769
tested cell models (Extended Data Fig. 5a). **b**, RNA sequencing analysis and GSEA
showing changes in expression of the 'RAC1_GTPase_Cycle' gene set in MDA231-
Cas9+ cells transduced with sgCtrl and sgITGAV for 3 days ($n = 3$ independent
sgRNA sequences per group). **c**, Western blot of RAC1 and β-actin in MDA231-
Cas9+ cells transduced with sgCtrl ($n = 2$ independent sgRNA sequences) and
sgRAC1 for 3 days ($n = 3$ independent sgRNA sequences). **d**, Growth competition
assay of MDA231-Cas9+ cells transduced with RFP-labeled sgCtrl (gray lines; two
independent sgRNA sequences) and sgRAC1 (purple lines; three independent

sgRNA sequences). Asterisk indicates that all three sgRAC1 groups were
significantly different ($P < 0.01$) from the two sgCtrl groups ($n = 3$ for each
group). **e,f**, Cellular apoptosis as detected by Annexin V+/DAPI− (**e**) and cell cycle
monitored by EdU incorporation (**f**) in MDA231-Cas9+ cells transduced with sgCtrl
and sgRAC1 for 3 days ($n = 3$ for each group). **g**, Representative fluorescence
images of F-actin (fluorescein isothiocyanate (FITC), green) and nucleus (DAPI,
blue) staining in MDA231-Cas9+ cells transduced with sgCtrl, sgITGAV and
sgRAC1. Scale bars, 20 μm. **h**, Violin plot showing distribution of cell size (μm²)
in MDA231-Cas9+ cells transduced with sgCtrl, sgITGAV and sgRAC1. Data are
presented as the mean ± s.e.m. $P$ values were calculated by two-sided Student's
*t*-test. NS, not significant.

energy transfer (NanoBRET) assay (Fig. 4e)[39,40] by fusing the energy
donor NanoLuc luciferase to the cytoplasmic tail of ITGB5. We also
fused the energy acceptor 'HaloTag' to the cytoplasmic tail of ITGAV.
When adding the NanoLuc luciferase substrate (generating a 460 nm
donor signal) and the HaloTag ligand (generating a 618 nm acceptor
signal), this integrin αVβ5 reporter will turn on the NanoBRET signal
(618 nm versus 460 nm ratio) while the ITGAV and ITGB5 subunits are
proximal (Fig. 4e, left). Conversely, the NanoBRET signal will be abol-
ished upon disengagement between the ITGAV and ITGB5 subunits
(Fig. 4e, right). Alanine substitution of any one of the β-propeller HIP
residues (F51A, W123A, F189A, Y254A, F308A, S372A or Y436A) on
ITGAV attenuated the NanoBRET signal by 50% to 70% compared with
wild-type ITGAV (Fig. 4f; alanine substitution of the aromatic residues
outside of HIP are shown in Extended Data Fig. 6), indicating a pivotal
role of the ITGAV β-propeller HIP in integrin αVβ5 assembly.

### A lead compound that disrupts ITGAV heterodimerization
To identify additional classes of inhibitors that block ITGAV heter-
odimerization, we developed a CRISPR-TICA pipeline that enables
de novo identification of small molecular compounds for binding to
CRISPR-hypersensitive surface areas of the targeted protein. We rea-
soned that the CRISPR-hypersensitive surface areas (which cannot
tolerate CRISPR-induced mutagenesis; Fig. 4d) might indicate criti-
cal functional positions amenable to pharmaceutical inhibition. We
mapped the NCS on a crystal structure of the ITGAV β-propeller domain
(Fig. 5a; PDB ID: 3IJE)[41–43] and used AutoSite[44] and AutoDock Vina[45] to
predict the binding affinity of ~128 K diverse compounds (collected

by the National Cancer Institute/Developmental Therapeutics Pro-
gram (NCI/DTP) Open Chemicals Repository) to the β-propeller HIP
(Fig. 5a, box). We then selected the top 500 predicted binders (Fig. 5b
and Supplementary Table 6; binding free energies ($\Delta G°$) ≤ −11.6 kJ mol⁻¹)
for CellTiter Glo and CCK8 (Cell Counting Kit 8) validation screens in
MDA231 cells (Fig. 5c and Supplementary Table 7; each compound
tested at 10 μM) and identified nine candidate compounds that could
suppress the viability of MDA231 cells with <10% viability in both assays
(Fig. 5d and Extended Data Fig. 7).

As mutagenesis of the β-propeller HIP of ITGAV significantly
affected the assembly of integrin αVβ5 (Fig. 4f), we utilized a flow cyto-
metric assay to detect integrin αVβ5 dimers on the cell surface using
an monoclonal anti-integrin αVβ5 heterodimer antibody (Fig. 5e and
Extended Data Fig. 3b). This allowed us to monitor the rapid changes in
integrin αVβ5 levels on the cell surface upon 1 h compound treatments.
We found that most of the candidate compounds were unable to reduce
cell surface integrin αVβ5 levels (Cpd_AV5/82/259/343/377/388/469) or
exhibited a notable fluorescent background (Cpd_AV84), suggesting
that the cell inhibitory effects of these compounds were not associated
with integrin αVβ5 disruption (Fig. 5f, green). On the other hand, one
of the nine candidate compounds (Cpd_AV2) exhibited potential to
reduce the presence of integrin αVβ5 on the cell surface (Fig. 5f, orange;
minimum inhibitory concentration ~40 μM with half-maximal inhibi-
tory concentration ($IC_{50}$) ~6.9 μM; additional validation data shown
in Supplementary Fig. 3). At the cellular level, we found that Cpd_AV2
led to pronounced apoptosis and cell cycle arrest by 3 h posttreatment
(Fig. 5g, h). Furthermore, Cpd_AV2 induced a drastic change in cell

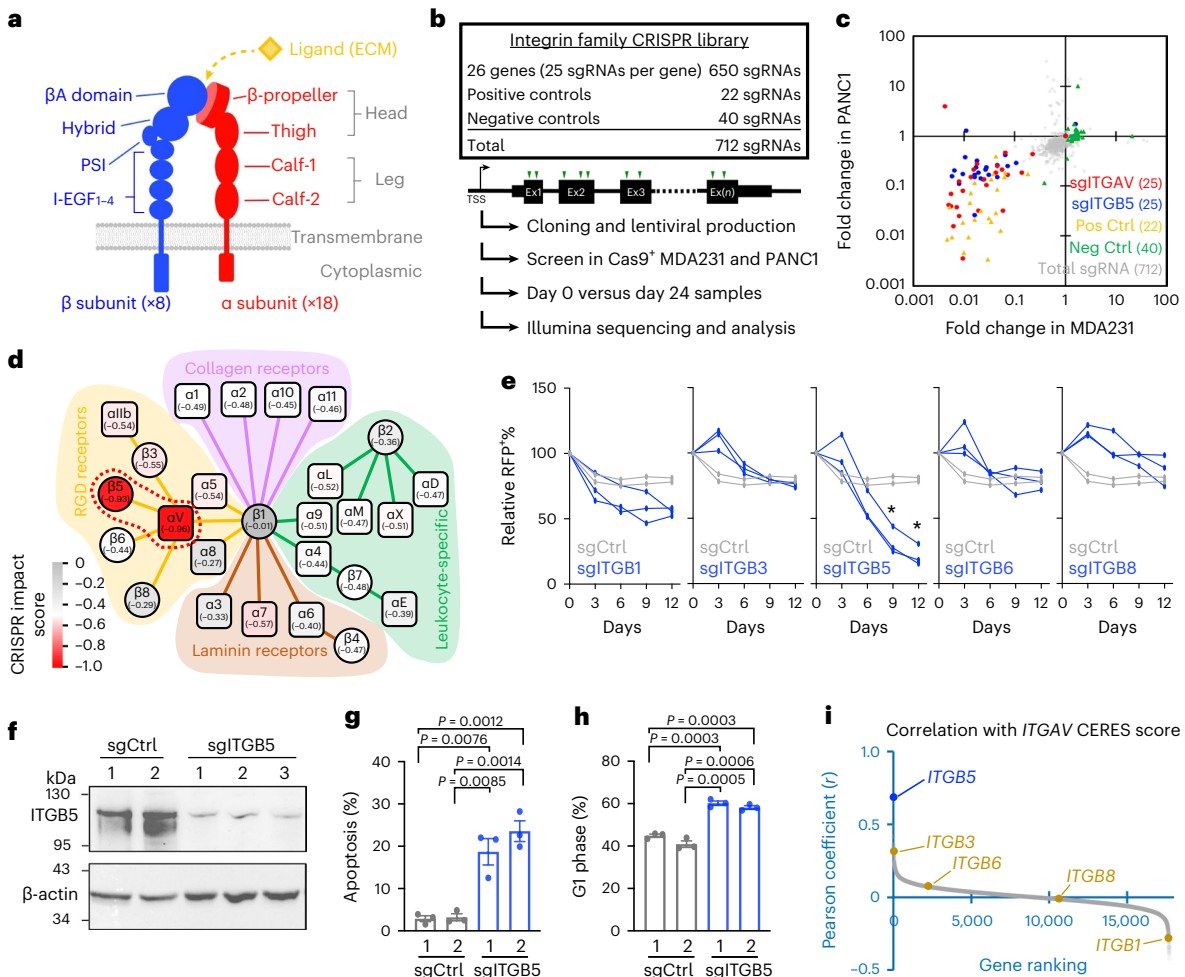

**Fig. 3 | Integrin family CRISPR screens reveal the critical role of integrin αVβ5 in cancer cell expansion. a**, Model of integrin α (red) and β (blue) subunits and domain structures. The binding site of the extracellular ligand (yellow) is assembled upon heterodimerization of the α/β subunits. **b**, Schematic outline of integrin family CRISPR screens (712 sgRNAs) in Cas9-expressing MDA231 and PANC1 cells. **c**, Fold change of each sgRNA from day 0 to day 24 in MDA231-Cas9+ (x axis) and PANC1-Cas9+ (y axis) cells. The sgRNAs targeting *ITGAV* (red dots), *ITGB5* (blue dots), positive controls (yellow triangles), negative controls (green triangles) and the total library (gray dots) are indicated. **d**, Heatmap showing CRISPR impact scores (median $\log_{10}$ fold change of 25 sgRNAs) of each integrin subunit in the integrin network consisting of 24 distinct integrin α/β heterodimers. The solid lines indicate the integrin α/β pairs forming the RGD receptors (yellow), collagen receptors (pink), laminin receptors (brown) and leukocyte-specific receptors (green). The red dotted circle highlights αVβ5 as the top essential integrin heterodimer in cancer cells. **e**, Growth competition

assay of MDA231-Cas9+ cells transduced with RFP-labeled sgCtrl (gray lines; two independent sgRNA sequences) and sgITGB1/3/5/6/8 (blue lines; three independent sgRNA sequences for each gene). Asterisk indicates that all three sgRNAs for each ITGB gene group were significantly different ($P < 0.01$) from the two sgCtrl groups ($n = 3$ for each group). **f**, Western blot of ITGB5 and β-actin in MDA231-Cas9+ cells transduced with sgCtrl ($n = 2$ independent sgRNA sequences) and sgITGB5 ($n = 3$ independent sgRNA sequences) for 3 days. **g,h**, Cellular apoptosis detected by Annexin V+/DAPI− (**g**) and cell cycle monitored by EdU incorporation (**h**) in MDA231-Cas9+ cells transduced with sgCtrl and sgITGB5 for 3 days ($n = 3$ for each group). **i**, Gene ranking based on the Pearson coefficient (r) of CERES scores between *ITGAV* and *ITGB5* (blue) compared with other *ITGAV* partner β subunit genes *ITGB1/3/6/8* (yellow) in the 769 tested cell models (Extended Data Fig. 5b). Data are presented as the mean ± s.e.m. P values were calculated by two-sided Student's t-test.

morphology, cytoskeleton assembly and cell size as early as 10 min posttreatment (Fig. 5i, j). These effects of Cpd_AV2 resembled those triggered by ITGAV depletion (Figs. 1f,g and 2g,h), marking Cpd_AV2 as the top integrin αVβ5 disruptor from the compound screen.

To validate the interaction between Cpd_AV2 and ITGAV, we purified the recombinant His⁶-tagged ITGAV [31–492 amino acids (aa)] from *Escherichia coli* (Fig. 6a; covers the β-propeller domain of ITGAV) and examined the protein thermal stability under control versus Cpd_AV2 conditions. We observed that incubation of Cpd_AV2 (40 μM) increased the melting temperature ($T_m$) from 52.0 °C to 58.1 °C (Fig. 6b; $\Delta T_m$ = 6.1 °C), suggesting an interaction between Cpd_AV2 and the purified ITGAV β-propeller domain. Furthermore, molecular docking of the bound complexes demonstrated favorable interactions between Cpd_AV2 (yellow) and the ITGAV β-propeller central cavity

(Fig. 6c) with a superior binding energy ($\Delta G°$ = −15.0 kJ mol⁻¹; ranked in the top two of the ~128 K docked compounds), showing competitive binding of Cpd_AV2 against the ITGB5 βA loop (right panel, cyan fragment; specifically at the K287 position) on the surface of the ITGAV β-propeller domain. Molecular dynamics simulations starting from the AlphaFold2 (ref. 35) structural model of the ITGAV–ITGB5 complex showed that K287 of ITGB5 interacts closely with multiple aromatic residues in ITGAV (Extended Data Fig. 8a–d), thereby playing a critical part in holding the complex together. This interaction is broken by the presence of Cpd_AV2 (Extended Data Fig. 8e). Mutation of K287 to alanine significantly attenuated the ITGAV–ITGB5 Nano-BRET signal (Extended Data Fig. 8f), highlighting an essential role of the ITGB5 K287 in integrin αVβ5 assembly. These analyses together provide mechanistic insights into the role of Cpd_AV2 in disrupting

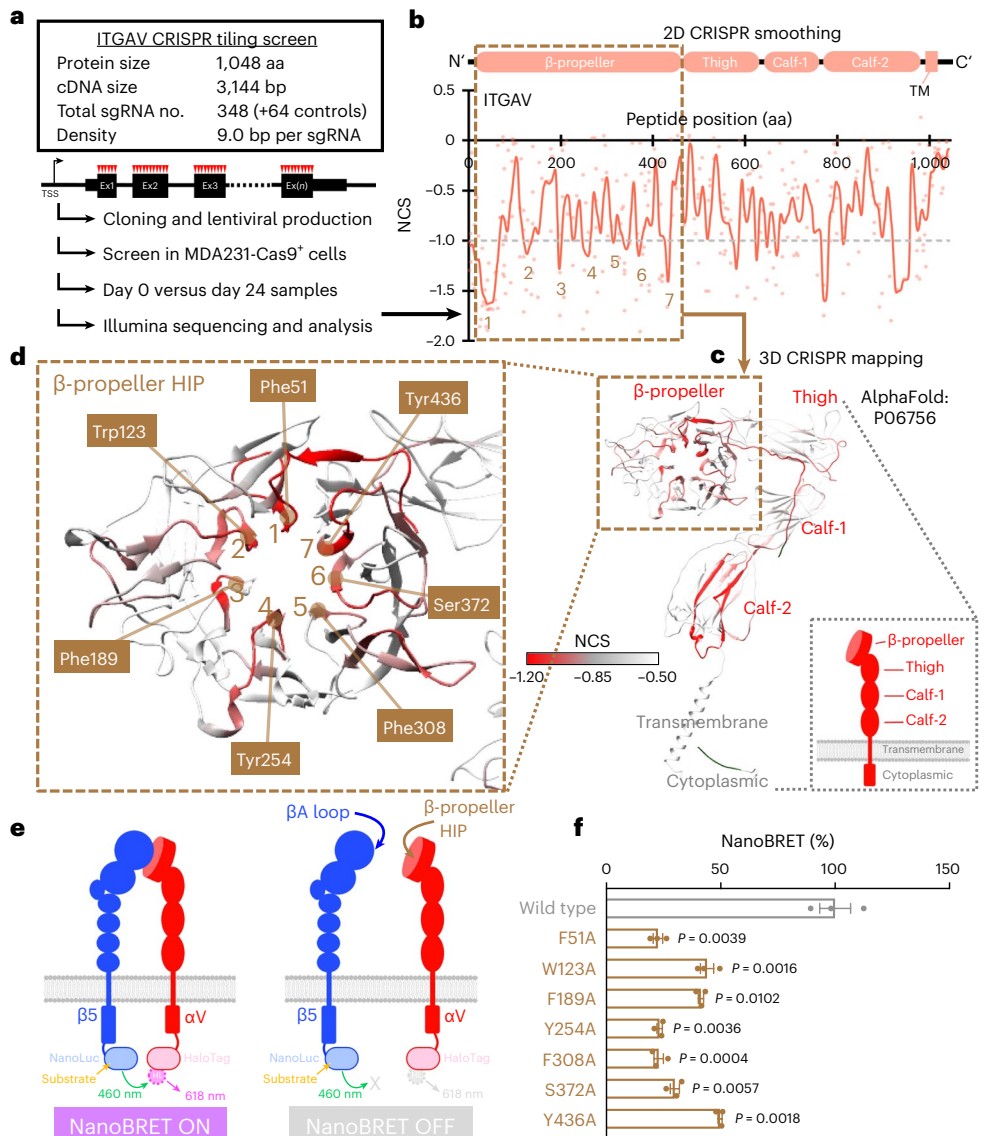

**Fig. 4 | High-density CRISPR tiling identifies a critical pocket in the β-propeller domain of ITGAV. a**, Schematic outline of the ITGAV high-density CRISPR-tiling scan (412 sgRNAs) in MDA231-Cas9⁺ cells. **b**, 2D annotation of ITGAV CRISPR scan. The red line indicates the smoothened model of NCS derived from 348 sgRNAs (dots) targeting the coding exons of *ITGAV*. The median NCS of the positive control (gray dotted line; defined as −1.0) and negative control (defined as 0) sgRNAs are highlighted. The brown dashed box contains the β-propeller domain. The numbers 1–7 pinpoint the CRISPR-hypersensitive regions within the β-propeller domain. **c**, 3D annotation of ITGAV CRISPR scan NCS relative to AlphaFold structural modeling of ITGAV (AlphaFold ID: P06756).

**d**, Enlarged view of the β-propeller domain showing the CRISPR-hypersensitive regions (numbers 1–7 as indicated in **b**) pointing to the center cavity of the β-propeller HIP. The residues contributing to this aromatic-enriched pocket are highlighted. **e**, Schematic outline of the NanoBRET reporter system for detecting the ITGAV–ITGB5 interaction in living cells. **f**, Effect of alanine substitution of the ITGAV β-propeller HIP residues (brown; *n* = 3 for each group) on the NanoBRET signal compared with the wild-type ITGAV (gray; *n* = 3 for each group). Data are represented as mean ± s.e.m. *P* values were calculated by two-sided Student's *t*-test. Ex, exon; TSS, transcription start site. TM, transmembrane.

the ITGAV–ITGB5 interaction. Compared with Cpd_AV2, we found that cilengitide[46–48] (an RGD-mimetic ITGAV inhibitor examined in multiple clinical trials) was unable to disrupt the level of cell surface integrin αVβ5 heterodimer (Fig. 6d) and resulted in less efficacy of cell suppression (Fig. 6e). Investigation of the effectiveness of Cpd_AV2 against commonly used cancer cell models indicated a utility of Cpd_AV2 for pan-cancer treatment (Fig. 6f; IC₅₀ = 1.3–5.2 μM in cell cultures). Collectively, our data suggest that Cpd_AV2 (NCI/DTP NSC identifier: 268394; IUPAC name: (5S,7R,12S,14S)-5,14-dimethyl-7,12-dinaphthalen-2-yl-1,4,8,11-tetrazacyclotetradecane; structure shown in Fig. 6g) acts through disrupting the ITGAV heterodimerization to eliminate integrin αVβ5 signaling, providing an alternative and potentially more potent ITGAV-targeted therapeutic approach compared with traditional RGD-blockers (Fig. 6h).

## Discussion

The cell surface proteome is enriched for structural and signaling components that mediate diverse biological activities under normal and disease conditions[1–4]. A better understanding of the cell surface protein genes related to cancer progression could provide new therapeutic opportunities and shed light on novel mechanisms of drug action. In this study, we performed a series of CRISPR genetic screens (that is, a cell surface proteome screen, an integrin family screen and a high-density ITGAV CRISPR-tiling screen) in multiple cancer cell

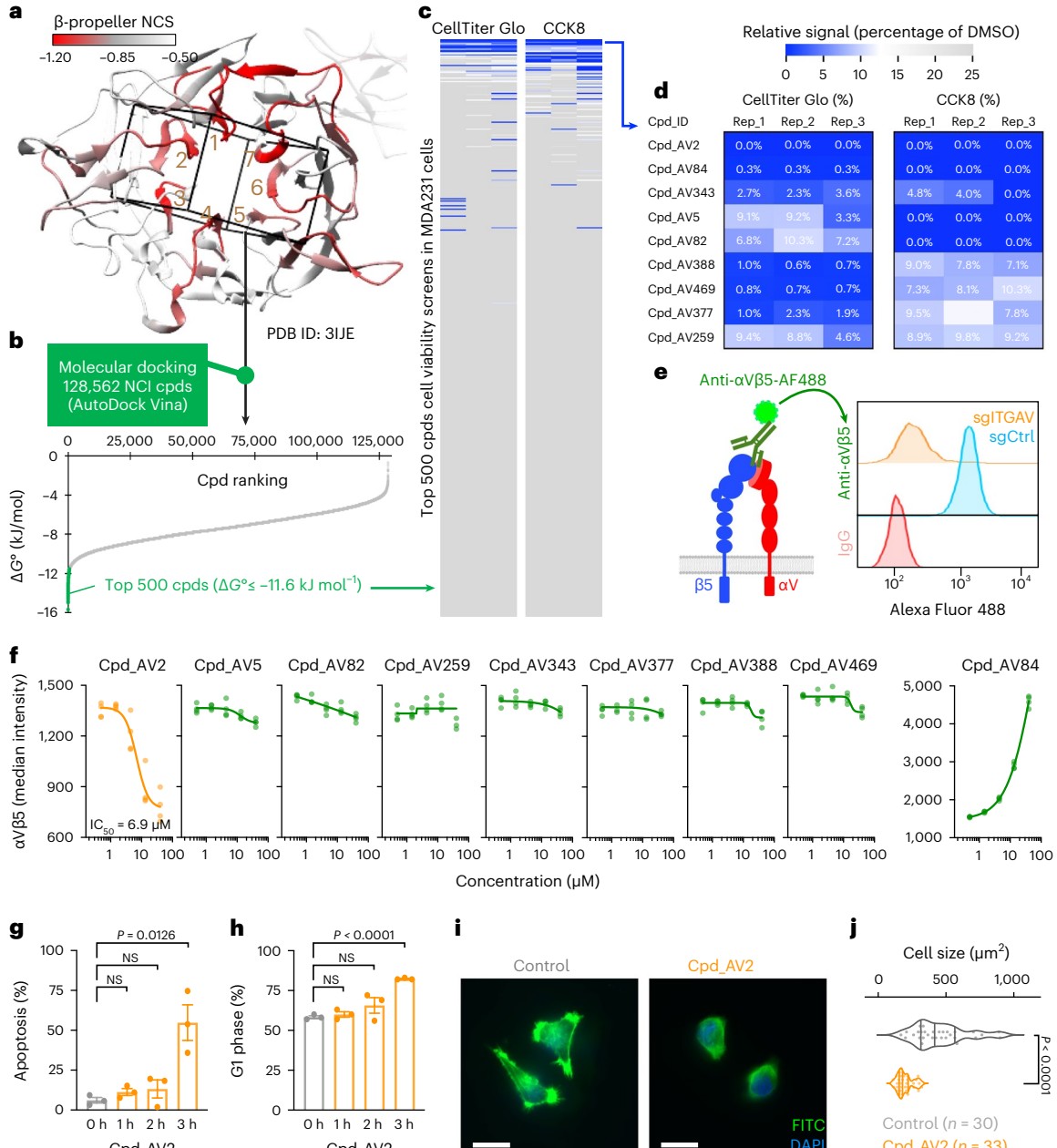

**Fig. 5 | Identification of compounds targeting ITGAV β-propeller domain by CRISPR-TICA pipeline. a**, 3D 'docking box' (cube) defined by the CRISPR-hypersensitive regions (numbers 1–7) within the ITGAV β-propeller domain. **b**, Compound (Cpd) ranking based on free binding energy (ΔG°) to the 'docking box' within the β-propeller domain predicted by AutoDock Vina. **c,d**, Heatmap showing relative CellTiter Glo (left) and CCK8 (right) signals (percentage of the signal for dimethyl sulfoxide; DMSO) in MDA231 cells incubated with 10 μM of 500 selected compounds (**c**) and the top nine effective compounds (**d**) for 3 days. Effective cell killing was defined as less than 10% relative signals for both CellTiter Glo and CCK8 assays. **e**, Schematic outline of flow cytometric measurement of cell surface integrin αVβ5 using a monoclonal antibody against integrin αVβ5

heterodimers. **f**, Effects of the top nine candidate compounds on cell surface integrin αVβ5 levels upon 1 h compound treatments (n = 4 for each condition). **g,h**, Cellular apoptosis detected by Annexin V⁺/DAPI⁻ (**g**) and cell cycle monitored by EdU incorporation (**h**) in MDA231 cells treated with Cpd_AV2 (40 μM) for 0 to 3 h (n = 3 for each time point). **i**, Representative fluorescence images of F-actin (FITC, green) and nucleus (DAPI, blue) staining in MDA231 cells treated with control (DMSO) and Cpd_AV2 (40 μM) for 10 min. Scale bars, 20 μm. **j**, Violin plot showing the distribution of cell size (μm²) in MDA231 cells treated with control (DMSO) and Cpd_AV2 (40 μM) for 10 min. Data are presented as the mean ± s.e.m. P values were calculated by two-sided Student's t-test.

models. Using these functional genetics approaches, we identified the critical role of integrin αVβ5 in cancer cell maintenance. We also demonstrated that the CRISPR-hypersensitive cavity within the β-propeller of ITGAV is indispensable to integrin αVβ5 heterodimerization, offering a therapeutic pocket for pharmacological development.

ITGAV (also known as CD51) is expressed in multiple tissue cell types and is involved in tissue developmental steps including

angiogenesis[49,50]. The ITGAV integrins (αVβ1, αVβ3, αVβ5, αVβ6, αVβ8) are heterodimeric cell surface receptors that recognize RGD-containing ECM proteins (for example, vitronectin, fibronectin, osteopontin, von Willebrand factor and thrombospondin)[6] and intercellular signaling molecules (for example, TGFβ)[51,52]. Serving as the primary receptors for vitronectin (offered to cells under standard cell culture conditions for cell-to-plate adhesion), integrins αVβ3 and

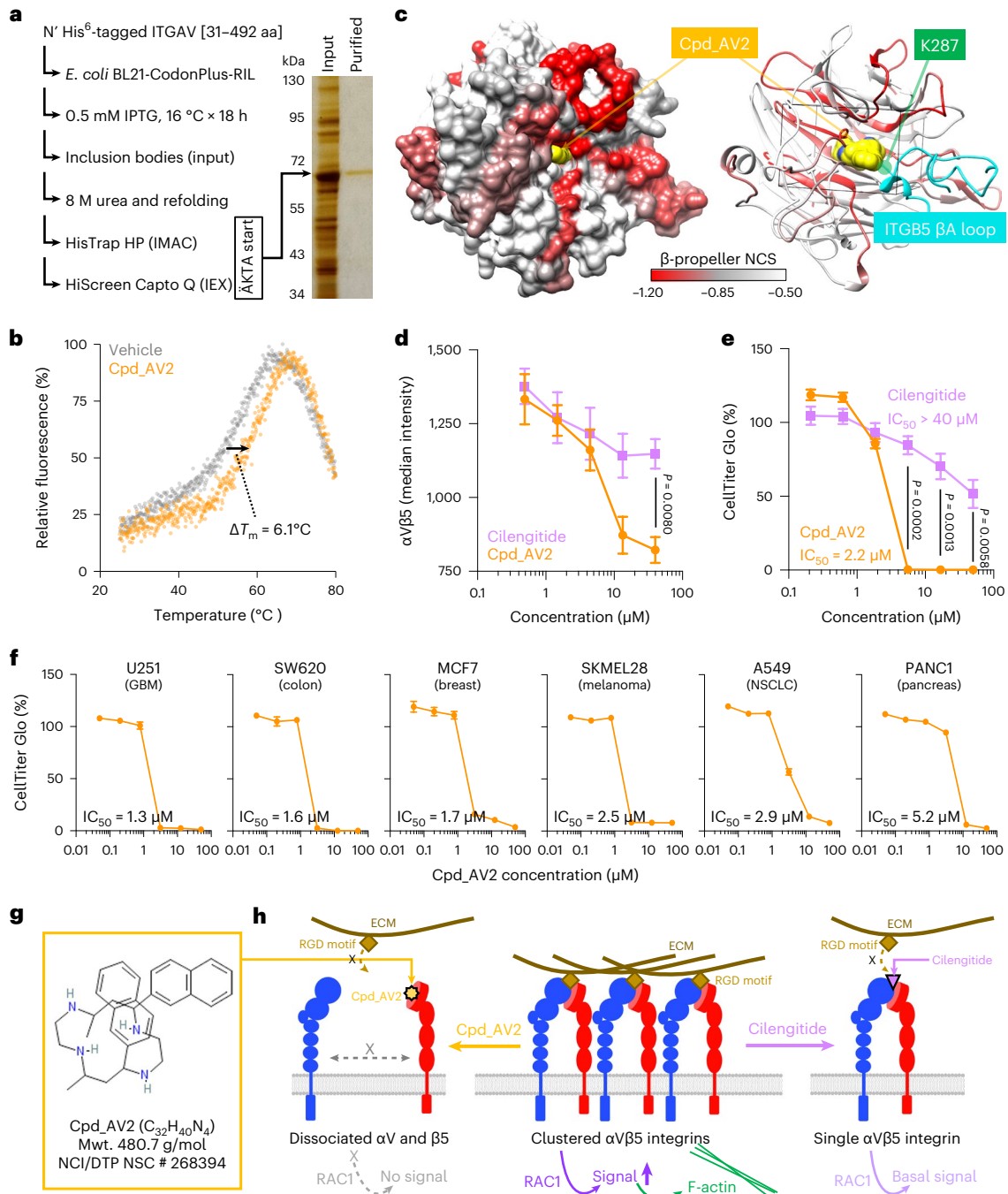

**Fig. 6 | Characterization of ITGAV β-propeller domain inhibitor Cpd_AV2.**
**a,** Purification of bacterial-expressed recombinant ITGAV β-propeller domain (peptide region 31–492 aa; N-terminal His[6]-tagged) using immobilized metal affinity chromatography (IMAC) and anion exchange chromatography (IEX). The input and purified ITGAV β-propeller domain samples were visualized by gel electrophoresis and silver staining (right; gel representative of two independent protein purification experiments). **b,** Protein thermal stability as estimated by fluorescent dye incorporation of the purified ITGAV β-propeller domain under control (DMSO) and Cpd_AV2 (40 μM) conditions. **c,** Protein surface model (left) showing a docking simulation of the ITGAV β-propeller domain (colored by NCS) interacting with Cpd_AV2 (yellow). Protein ribbon model (right) illustrates an overlap of ITGB5 lysine 287 (within βA loop; cyan) and Cpd_AV2 (yellow) binding on the β-propeller HIP of ITGAV. **d,e,** Effects of cilengitide and Cpd_AV2 on cell surface integrin αVβ5 levels after 1 h treatment (**d**) and cell expansion after 72 h treatment in MDA231 cells (**e**) (n = 3 for each group). **f,** Effects of 72 h Cpd_AV2 treatment on expansion of six cancer cell models (n = 3 for each group). **g,** Chemical structure of Cpd_AV2 (source: NCI/DTP Open Chemicals Repository). **h,** Model showing distinct mechanisms of action between Cpd_AV2 (left) and cilengitide (right) for suppressing ECM-to-integrin αVβ5 signaling (middle). Data are presented as the mean ± s.e.m. P values were calculated by two-sided Student's t-test. NSCLC, nonsmall-cell lung cancer.

αVβ5 are known to protect tumor-derived cells from apoptosis[53,54]. Increased expression of ITGAV has been reported as a prognostic marker in diverse cancer types (breast cancer, prostate cancer, ovarian cancer, glioblastoma, myeloma, hepatocellular carcinoma, skin carcinoma, colorectal adenocarcinoma, esophageal adenocarcinoma, pancreatic adenocarcinoma and so on)[9,10,48,55–62]. A recent study also demonstrated the role of integrin αVβ5 in Zika virus entry to glioblastoma stem cells[63]. Genetic suppression of ITGAV was shown to impair

the proliferation, survival and migration of cancer cells, suggesting that ITGAV could serve as a therapeutic target to inhibit tumor progression and metastasis[11]. Owing to the potential of ITGAV inhibition in the therapeutics market, pharmacological targeting of ITGAV integrins has been widely explored over the past 20 years, with more than 30 inhibitors currently under clinical and preclinical development[64]. Nonetheless, the efficacy of current ITGAV-targeted therapies remains elusive, emphasizing the need for a novel and perhaps more effective blockade for ITGAV signaling.

Thus far, ITGAV inhibitors have been primarily focused on blocking interactions between heterodimerized ITGAV integrins (for example, αVβ3 and αVβ5) and RGD-containing ECM proteins[9,10]. Specifically, the current ITGAV integrin inhibitors belong to two principal strategies: there are RGD-mimetic molecules (for example, cilengtide, GLPG0187 and MK-0429) and ITGAV-integrin-specific blocking antibodies (intetumumab, etaracizumab, abituzumab (EMD 525797) and so on)[11,64,65]. Although it prevents the anchorage of cancer cells to ECM proteins, the most advanced compound among these traditional integrin inhibitors (cilengitide) showed limited benefits in several animal and human trials[66–68], potentially owing to incomplete suppression of basal integrin signaling, as the α/β heterodimers remained undisrupted (Fig. 6h, right). On the other hand, no compounds that disrupt the interactions between ITGAV and its partner β subunits have previously been reported. Our results highlight a class of inhibitory mechanism that dissociates the integrin αVβ5 by blocking the CRISPR-hypersensitive β-propeller pocket (which cannot tolerate CRISPR-induced mutagenesis; Fig. 4d) in ITGAV. This strategy provides an additional and perhaps more effective therapeutic action by eliminating basal integrin heterodimer signaling (Fig. 6h, left). Furthermore, this drug action might also prevent inside-out integrin activation[6,8] from attenuating therapeutic efficacy.

Structurally, the basic amino acid encapsulated in the β-propeller of ITGAV is conserved across the ITGB1/3/5/6/8 peptides (Extended Data Fig. 9a; K/R287), highlighting the potential of Cpd_AV2 to disrupt the functions of αVβ1, αVβ3, αVβ5, αVβ6 and αVβ8 integrins. As a proof of concept, we monitored the integrin-αVβ6-dependent cell adhesion of HT-29 colorectal carcinoma cells to fibronectin reported by ref. 69. We found that preincubation of the HT-29 cells with Cpd_AV2 led to dose-dependent blockade of HT-29 cell adhesion to the fibronectin-coated wells (Extended Data Fig. 9b). We foresee that this integrin-targeting strategy will also be applicable to other integrin α subunits for compounds binding to their specific β-propeller pockets (a common feature within the integrin α subunits). Notably, the function of integrins is heavily influenced by the glycosylation of their extracellular domains[70]. In addition to the in vitro protein/compound biochemical assays that utilize nonglycosylated bacterially expressed proteins (Fig. 6a,b), cell-based characterizations such as cell surface flow cytometry and the NanoBRET interaction assay (Fig. 5f and Supplementary Fig. 3) in mammalian cells are necessary to validate the impact of compounds on the full-length glycosylated integrins.

High-throughput CRISPR library screens have been performed in diverse cancer cell types, revealing critical mechanisms mediating tumorigenesis and therapeutic response[15–19]. By contrast, the potential of CRISPR technology to investigate gene function at subgene (that is, protein domain or motif) resolution is now being explored[20–22]. For example, high-resolution CRISPR gene tiling screens have been used to identify the essential elements within catalytic core domains[22,24,26]. In addition, CRISPR tiling has the sensitivity to pinpoint protein–protein interaction sites within screened proteins[21,25,28]. Our CRISPR gene scan has also been used to identify a protein domain mediating oncoprotein nuclear trafficking[27]. In the present study, we further exploited the utility of high-density CRISPR gene tiling to identify a protein surface pocket for therapeutic development (Figs. 4 and 5). We propose that CRISPR-hypersensitive surface areas (that is, those that cannot tolerate CRISPR-induced mutagenesis) might correspond to critical positions

that, when targeted by small molecules, could disrupt the normal function of the protein. To validate this hypothesis, we obtained previously published CRISPR-tiling data from ref. 23 and identified four proteins with well-defined inhibitors targeting their CRISPR-hypersensitive pockets (Extended Data Fig. 10; including the bromodomain of BRD4 and the kinase catalytic cores of AURKB, CDK1 and WEE1)[71–74]. These analyses demonstrated the utility of CRISPR tiling as a generalizable approach for future drug discovery.

In short, our ITGAV CRISPR-tiling scan offered ~3.0 aa per sgRNA resolution (Fig. 4a, b) and clearly distinguished the CRISPR-hypersensitive central pocket from the surrounding β-propeller domain (Fig. 4d). Notably, this 3D pocket was assembled from seven discontinuous CRISPR-hypersensitive segments (Fig. 4b, labeled 1–7; separate from each other in their two-dimensional (2D) peptide positions) in the ITGAV β-propeller domain, highlighting the capacity of the CRISPR-tiling scan for subdomain functional recognition beyond traditional domain mapping. This finding prompted us to develop the CRISPR-TICA workflow for de novo compound discovery and enabled us to identify a lead inhibitor (Cpd_AV2) that disrupts integrin heterodimerization.

## Availability of materials

Cas9-expressing cells; ITGAV cDNA; and the CRISPR library for the cell surface proteome (2,973 sgRNAs), integrin family (714 sgRNAs) and ITGAV gene tiling scan (412 sgRNAs) are available upon request. All other biological materials are commercially available.

### Reporting summary

Further information on research design is available in the Nature Portfolio Reporting Summary linked to this article.

## Online content

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

## Methods

### Cell lines and cell culture

HEK293, PANC1 and SW620 cells were obtained from the American Type Culture Collection; MDA231 (that is, MDA-MB-231) cells were obtained from M. Feng (City of Hope Cancer Center); H661 cells were obtained from J. Qi (Dana Farber Cancer Institute); and U251 cells were obtained from M. Chen (City of Hope Cancer Center). Cells were cultured in Dulbecco's modified Eagle medium (DMEM; Gibco) supplemented with 10% fetal bovine serum (Omega Scientific). All media were supplemented with penicillin (100 units per ml; Gibco), streptomycin (100 µg ml⁻¹; Gibco), L-glutamine (2 mM; Gibco) and plasmocin (0.5 µg ml⁻¹; InvivoGen). All cells were cultured in a 37 °C incubator with 5% $CO_2$. Cells stably expressing the Cas9 endonuclease were established via transduction of LentiCas9-Blast (52962, Addgene) lentivirus and selected using blasticidin (Gibco). For assay preparation, all adherent cells were removed from plates with a nonenzymatic cell dissociation buffer (13150016, Gibco).

### CRISPR library and single sgRNA cloning

Briefly, guide RNA oligos were synthesized by microarray (CustomArray; for library cloning) or individual oligosynthesis (IDT; for single sgRNA) and cloned into the ipUSEPR lentiviral sgRNA vector (hU6-driven sgRNA coexpressed with EF-1a-driven red fluorescent protein (RFP) and puromycin-resistance gene) using the *Bsm*BI (NEB) restriction sites (Extended Data Fig. 1a). CRISPR sgRNAs were selected using the BROAD Institute Genetic Perturbation Platform – CRISPick[17]. For the cell surface proteome CRISPR library, 2,905 sgRNA sequences targeting 581 genes encoding cell surface proteins were designed (Extended Data Fig. 1b; five sgRNAs per gene). For the integrin family CRISPR library, 650 sgRNA sequences targeting 26 genes encoding integrin subunits were selected (Extended Data Fig. 1c; 25 sgRNAs per gene). For the ITGAV tiling scan CRISPR library, 348 sgRNA sequences targeting every protospacer adjacent motif within the human *ITGAV* coding exons were covered (Extended Data Fig. 1d; 9 bp per sgRNA). The cloned libraries were first sequenced with a NextSeq to ensure at least 90% of the sgRNA sequences exhibited a minimal ten reads per million reads. Quality control sequencing reports for these libraries are shown in Supplementary Table 2. The sequences of single sgRNAs selected for validation experiments are listed in Supplementary Table 8.

### Lentiviral production and transduction

Lentiviruses were produced in HEK293 cells (CRL-1573, American Type Culture Collection) with packaging plasmids pPAX2 (12260, Addgene) and pMD2.G (12259, Addgene). Then, pPAX2, pMD2.G and a lentiviral backbone plasmid were mixed in a 1:1:1 ratio in Opti-MEM medium (31-985-062, Gibco) in the presence of 50 µg ml⁻¹ polyethyleneimine (PRIME-P100-100MG, Serochem LLC). Twenty-four hours after transfection of HEK293 cells, the medium supernatant was aspirated and replaced with fresh DMEM. Then, the transfected cells were allowed to grow for 48 h to produce lentiviruses. Subsequently, the virus-containing supernatants were incubated with 10% polyethylene glycol (BP233-1, ThermoFisher Scientific) at 4 °C overnight and then centrifuged at 3,000g, 4 °C, 30 min, to collect precipitated viral particles. After that, the viral pellets were resuspended with appropriate DMEM, aliquoted and kept at −80 °C.

### CRISPR library screens

The CRISPR library screens were performed as previously described[28]. Briefly, the CRISPR sgRNA libraries were delivered to Cas9-expressing cells using lentiviral infection (~15% transduction rate, monitored based on RFP expression). To achieve 1,000× coverage of the library in each screen, 30 million cells for the cell surface proteome library screen, six million cells for the integrin family library screen and four million cells for the ITGAV high-density CRISPR-tiling scan were used to start each screen replicate. The library-infected cells were then selected

using puromycin (1.5 µg ml⁻¹; Gibco) and subcultured every 3 days. The integrated sgRNA at the start (day 0) and end (day 24) timepoints was amplified by PCR (NEBNext Ultra II Q5; NEB) using the previously reported DCF01 5′-CTTGTGGAAAGGACGAAACACCG-3′ and DCR03 5′-CCTAGGAACAGCGGTTTAAAAAAGC-3′ primers[22]. After sequencing with a NextSeq550 (Illumina), the read count of each 20 nucleotide sequence that matched an sgRNA in the library of guide RNA sequences was calculated. For the cell surface proteome screen, essential genes were identified using MAGeCK analysis[30]. For the integrin family gene panel screen, the CRISPR impact score was defined as the median log₁₀ fold change of the 25 sgRNAs for each integrin subunit encoding gene. For the ITGAV CRISPR gene tiling scan, the NCS indicated the frequency change of each sgRNA between the start and end of the screen on a log₁₀ scale, where the median score of the negative control sgRNA (defined as 0; sgRNA targeting nonessential sequences) and the median score of the positive control sgRNA (defined as −1.0; sgRNA targeting *MYC*, *BRD4*, *RPA3*, *PCNA* and so on) were obtained from the control sgRNAs within the screen libraries. Low-frequency sgRNAs (below 5% of the expected frequency) in the library were removed from the analysis.

### Annotation of CRISPR gene tiling scan

The ITGAV CRISPR gene tiling scan library (348 sgRNAs targeting *ITGAV* coding exons) was delivered into the MDA231-Cas9⁺ cells and processed using the methods described above. For 2D annotation, the NCSs of individual sgRNAs were processed by Gaussian kernel smoothing in R[22], and the average score over the trinucleotide codons was calculated for each peptide position. For 3D annotation, we first obtained 3D structure data for ITGAV from the AlphaFold Protein Structure Database (Protein ID: P06756)[35] and the Research Collaboratory for Structural Bioinformatics Protein Data Bank (RCSB PDB ID: 3IJE)[41–43]. Subsequently, the smoothened ITGAV CRISPR NCSs (from the 2D annotation) were mapped onto the ITGAV 3D structures using the 'Defined Attribute' and 'Render by Attribute' functions in UCSF Chimera 1.15 (ref. 75).

### CRISPR-TICA workflow

The human ITGAV β-propeller structure was extracted from 3IJE using PyMOL v.2.0.4 (Schrödinger, LLC) and the PDB 2PQR server[76], and the resultant pqr file was converted into pdbqt format using AutoDock-Tools[77]. The space within the CRISPR-hypersensitive region suitable for compound binding (the docking box shown in Fig. 5a) was suggested by AutoSite[44]. The 3D chemical structure of ~128 K diverse compounds (collected in the NCI/DTP Open Chemicals Repository; https://dtp.cancer.gov) downloaded as mol2 files were split into subsets of 20,000 compounds using Open Babel v.2.4.1 (ref. 78). Subsequently, each subset was converted into pdbqt format (the input file format for AutoDock Vina) using PyRx v.0.9.7 (ref. 79). Having prepared both ligand and protein structures for structure-based drug discovery, we used AutoDock Vina v.1.1.2 (ref. 45), an in silico molecular docking program, to virtually dock these compounds into the defined docking box using the City of Hope Saturn 2 Linux cluster. Finally, the docking data were processed and exported to csv files using Raccoon2 (ref. 77).

### Cell-based survival screen using CellTiter Glo and CCK8 assays

The top 500 ITGAV β-propeller binders suggested by CRISPR-TICA were requested from the NCI/DTP Open Chemicals Repository for functional validation. Compound information is listed in Supplementary Table 6. MDA231 cells were seeded at 10,000 cells per well in 96-well plates for 24 h, and the compounds were added to a final concentration of 10 µM for another 72 h. For the Cell Counting Kit 8 (CCK8) assay, 10 µl of CCK8 reagent (K1018, APExBio) was added to cells (100 µl per well), followed by incubation at 37 °C for 1 h, and the absorbance at 450 nm was measured using an Infinite M1000 Pro plate reader (Tecan Trading AG). For the CellTiter Glo assay, cells were washed twice with phosphate-buffered saline (PBS) and resuspended by trypsinization.

The resuspended cells (50 µl) were mixed with CellTiter Glo 2.0 reagent (10 µl; G9241, Promega) in white flat-bottomed 96-well plates (353296, Corning) at room temperature for 10 min, and the luminescence was measured using an Infinite M1000 Pro plate reader (Tecan Trading AG). The relative CellTiter Glo (%) and CCK8 (%) signals were normalized to the control condition (DMSO).

## Western blotting

Cells were harvested and lysed in 1% sodium dodecyl sulfate (SDS) lysis buffer (1% SDS, 50 mM Tris pH 7.5), and the proteins were denatured at 95 °C for 15 min. Protein concentration was measured using a DC Protein Assay Kit II (5000112, Bio-Rad). Denatured protein samples were separated on Bolt 4–12% Bis-Tris plus gels (NW04125, Invitrogen) or 3–8% Tris-acetate gels (EA0375, Invitrogen) using electrophoresis. The separated protein bands were transferred onto polyvinylidene fluoride (PVDF) Mini Stacks (0.2-µm pore size; IB24002, Invitrogen) using an iBlot 2 (Invitrogen). PVDF membranes were blocked with 5% bovine serum albumin (Fisher Scientific) in Tris-buffered saline with Tween-20 (TBST) at room temperature for 1 h and then probed with primary antibodies for ITGAV (4711, Cell Signaling Technology, 1:1000), ITGB5 (3629, Cell Signaling Technology; 1:1000), RAC1 (4651, Cell Signaling Technology; 1:1000) and β-actin (ab8226, Abcam; 1:5000) at 4 °C overnight. After the membranes had been washed with TBST three times, HRP-conjugated goat anti-mouse (31430, Invitrogen; 1:10,000) or goat anti-rabbit (31460, Invitrogen; 1:10,000) IgG secondary antibodies were added, followed by shaking at room temperature for 1 h. The washed PVDF membranes were then incubated with SuperSignal West Femto Substrate (P134095, ThermoFisher), and the chemiluminescence signals were detected using a ChemiDoc imaging system (Bio-Rad). The antibody concentrations are listed in Supplementary Table 10. The uncropped gel blot images are shown in Supplementary Fig. 4.

## Flow cytometric assays

Flow cytometric data were collected on an Attune NxT flow cytometer with an autosampler (ThermoFisher Scientific). For the RFP-coupled growth competition assay, the ipUSEPR vector system, which expresses an sgRNA together with a TagRFP fluorescent protein, was used to infect the Cas9+ cells at a ~50% transduction rate. The percentage of cells with an RFP fluorescence signal (RFP+%) was normalized to the RFP+% on day 0 (48 h after lentiviral infection). The cell cycle was monitored by EdU incorporation (10 µM EdU at 37 °C for 2 h) using Click-iT Plus EdU Alexa Fluor 647 Assay Kits (C10634, Invitrogen). Apoptotic cells were detected based on the Annexin V+/DAPI− population using an Annexin V Apoptosis Detection Kit (50-112-9048, Invitrogen). Live cells were defined by exclusion of 4′,6-diamidino-2-phenylindole (DAPI; D1306, Invitrogen) DNA staining. Cell surface integrin αVβ5 was recognized by a mouse monoclonal anti-human αVβ5 antibody (clone P1F76; sc-13588, Santa Cruz Biotech; 1:200) and stained with AF488-conjugated donkey anti-mouse IgG (ab150105, Abcam) secondary antibody. Another mouse monoclonal anti-human αVβ5 antibody (clone P1F6; 920005, BioLegend; AF647-conjugated) was used to validate the αVβ5 flow cytometry results.

## NanoBRET assays

To clone the constructs for the NanoBRET assays (Fig. 4e)[39,40], wild-type *ITGAV* and *ITGB5* cDNAs were subcloned from the open reading frame clones (HG11269 for *ITGAV* and HG10779 for *ITGB5*, Sino Biological) into the NanoBRET HaloTag and NanoLuc plasmids (N1821, Promega), respectively. Alanine substitution of the ITGAV β-propeller HIP (Fig. 4d,f; mutagenesis primers listed in Supplemental Table 9) was established using a Q5 site-directed mutagenesis kit (E0554S, New England Biolabs). All molecular cloning was performed with NEB 5-α competent *E. coli* cells (C2987; New England Biolabs). The final plasmids were validated via Sanger sequencing (Eton Bioscience). The NanoBRET HaloTag (ITGAV) and NanoLuc (ITGB5) plasmids were cotransfected

into HEK293 cells using FuGENE HD (E2311, Promega). The transfected cells were seeded at 20,000 cells per well in white 96-well tissue culture plates (353296, Falcon) for 24 h and incubated with 100 nM of HaloTag ligand (HaloTag NanoBRET 618 Ligand; generating a 618 nm acceptor signal; G980A, Promega). The wells without HaloTag ligand served as negative controls. When the NanoLuc luciferase substrate was added, the 460 nm donor signal and the 618 nm acceptor signal were measured with a Synergy Neo2 Reader (BioTek).

## Transcriptomic analysis

Total RNA from the sgCtrl- and sgITGAV-transduced cell samples was extracted using an RNeasy Mini Kit (74104, QIAGEN). The mRNA library prep was performed by Novogene Inc. and sequenced on a NovaSeq 6000 (Illumina) with ~20 million paired-end 150 bp reads per sample. We then mapped the raw sequence reads to the human GRCh38 genome using STAR v.2.6.1d. Raw counts were quantified using featureCounts v.1.6.4 and then normalized using the trimmed mean of M values method. The relative expression level of each gene was compared using the Bioconductor package 'edgeR.' In addition, GSEA (v.4.1.0) was used to evaluate gene pathways affected by sgITGAV.

## Purification of ITGAV β-propeller domain

**Recombinant protein expression.** To clone the pITGAV[31–492 aa] for expressing the recombinant β-propeller domain in *E. coli*, the full-length human ITGAV open reading frame clone (HG11269, Sino Biological) was PCR amplified (primers AV_BP_F: 5′-GAGAACCT GTACTTCCAATCCATGGAGTTCAACCTAGACGTGGACAG-3′ and AV_BP_R: 5′-GTCGACGGAGCTCGAATTCGGATCCTTAGAG CAGGTTTTATTGTCTTG-3′) and cloned into the pNIC28-Bsa4 vector (26103, Addgene), resulting an ITGAV β-propeller domain (residues Phe31 to Ser492; 50.3 kDa) sequence with an amino-terminal hexahistidine tag (His⁶-tag). For recombinant expression of the ITGAV β-propeller domain, the pITGAV[31–492 aa] plasmid was first transformed into *E. coli* (BL21-CodonPlus-RIL; 230240, Agilent Technologies) in the presence of 100 µg ml⁻¹ kanamycin and 50 µg ml⁻¹ chloramphenicol. The transformed *E. coli* was scaled up to 2 l liquid cultures in Terrific Broth (BP9728-500, ThermoFisher Scientific) at 25 °C until the optical density at 600 nm reached 0.8. Expression of the recombinant β-propeller domain was induced by adding 0.5 mM isopropyl-β-D-thiogalactopyranoside (BP1755-1, ThermoFisher) at 16 °C overnight. The *E. coli* pellet was collected by centrifugation (8,000*g*, 4 °C, 5 min) and sonicated (50% amplitude; 5 s bursts interrupted by 5 s pauses for 60 cycles) on ice in the presence of 500 U benzonase (70664, MilliporeSigma) and cOmplete Protease Inhibitor Cocktail (04693159001, Roche). The cell lysate was centrifuged at 10,000*g*, 4 °C, for 10 min, and the insoluble protein pellet (containing the recombinant protein inclusion body) was harvested for refolding and purification[80].

**Refolding and purification of the recombinant protein.** Briefly, the protein pellet was vortexed to resuspend it in Buffer B (10 mM Tris pH 8.0, 1% Triton X, 0.2 mM phenylmethylsulfonyl fluoride (PMSF)) and Buffer C (10 mM Tris pH 8.0, 0.2 mM PMSF), and the washed protein pellet was collected by centrifugation at 10,000*g*, 4 °C, for 15 min. This protein pellet was then dissociated in Buffer D (10 mM Tris pH 8.0, 8 M urea, 10 mM DTT, 0.2 mM PMSF) at 4 °C for 2 h and centrifuged at 30,000*g*, 4 °C, for 30 min. The supernatant was transferred to a prewet dialysis cassette (Slide-A-Lyzer G3, 10K molecular weight cut-off; A52973, ThermoFisher) and submerged in 500 ml of Buffer E (100 mM Tris pH 8.0, 3 M urea, 400 mM L-arginine monohydrochloride, 20 mM reduced L-glutathione, 2 mM oxidized L-glutathione) overnight. Next, the cassette was dialyzed in 2 l of Buffer A (10 mM Tris pH 8.0, 150 mM NaCl) at 4 °C for a total of 24 h (replaced with fresh Buffer A four times). The dialyzed sample (containing the refolded recombinant protein) was then centrifuged at 5,000*g*, 4 °C for 15 min and filtered through a

0.45 µm PES filter (124-0045, Thermo Scientific Nalgene). The clarified protein sample was further purified by immobilized metal affinity chromatography with a HisTrap HP column (95056-204, Cytiva) followed by anion exchange chromatography with a HiScreen Capto Q column (28926978, Cytiva) using an ÄKTA start protein purification system (GE Healthcare-Cytiva). The purified recombinant ITGAV β-propeller domain protein was checked by SDS polyacrylamide gel electrophoresis with silver staining and stored at −80 °C.

## Protein thermal shift assay
The concentration of the purified ITGAV β-propeller domain protein was measured with a DC Protein Assay Kit II (5000112, Bio-Rad). For each thermal shift reaction (50 µl each in 96-well plates), 2.5 µg purified ITGAV β-propeller domain protein was mixed with 40 µM (final concentration) of Cpd_AV2 (or with DMSO as the vehicle control) and Protein Thermal Shift Dye (final concentration preoptimized to 2×; 4462263, ThermoFisher). The protein melt reaction was performed using a QuantStudio 3 real-time PCR system (ThermoFisher) to examine the fluorescence signal (ROX channel) from 25 °C to 99 °C with a ramp rate of 0.1 °C s$^{-1}$. The protein melting temperature ($T_m$) and temperature shift ($\Delta T_m$) were calculated using JTSA (https://paulsbond.co.uk/jtsa).

## Data availability
The RNA sequencing data generated in this study are available via the Gene Expression Omnibus under accession GSE231339. All the data supporting the findings of this study are included in this article and its Supplementary Information. The 3D protein structure (PDB ID: 3IJE) was obtained from the Research Collaboratory for Structural Bioinformatics Protein Data Bank (https://www.rcsb.org). ITGAV expression data for breast, pancreas, brain, colon and lung cancers were obtained from the Gene Expression database of Normal and Tumor tissues (http://gent2.appex.kr/gent2/). Additional data that support the findings of this study are provided in the Supplementary Information. Source data are provided with this paper.

## Code availability
The computational code and tool packages used in this study include Genetic Perturbation Platform (BROAD Institute), Bowtie2 (Johns Hopkins University), UCSF Chimera 1.15 (UC San Francisco), Attune NxT v.3.1.2 (ThermoFisher), GSEA v.4.1.0 (UC San Diego and BROAD Institute), FASTQC v.0.11.8, Burrows-Wheeler Aligner v.0.7.17, MACS2 v.2.1.1, SAMtools v.1.10, STAR v.2.6.1d, featureCounts v.1.6.4, edgeR v.4.0.1, deepTools v.3.5.1, IGV 2.14.0 (BROAD Institute), PyMOL v.2.0.4 (Schrödinger, LLC), the PDB 2PQR server[76], AutoDockTools[77], AutoSite[44], Open Babel v.2.4.1 (ref. [78]), PyRx v.0.9.7 (ref. [79]), AutoDock Vina v.1.1.2 (ref. [45]), Raccoon2 (ref. [77]), JTSA (https://paulsbond.co.uk/jtsa), Bio-Rad ChemiDoc MP (Bio-Rad), MAGeCK v.0.5.9.2 (ref. [30]). IC$_{50}$ calculations and two-sided Student's $t$-tests were performed using Prism 9 (GraphPad). $P > 0.05$ was considered to indicate a nonsignificant result.

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

## Acknowledgements
We thank the NCI/DTP (https://dtp.cancer.gov) for providing the compound library. This work was supported by the American Society of Hematology (C.-W.C.); Alex's Lemonade Stand Foundation 18-11849 (C.-W.C.); Stand Up To Cancer & Cancer Research UK RT617 (C.-W.C.); and National Institutes of Health grants CA197489, CA233691, CA236626, CA243124, CA278050 (C.-W.C.), CA274649 (N.M.M.), CA233922 (S.T.R.), CA255250 and CA258778 (M.F.). The structural computational studies were supported by National Institutes of Health grant GM117923 (N.V.), and the sequencing, mass spectrometry and structural computational studies were supported by National Institutes of Health P30 award CA033572 (City of Hope Cancer Center). These funders were not involved in study design, data collection or analysis, the decision to publish or the preparation of the paper.

## Author contributions
N.M.M., A.K.N.C., K.M., W.-H.C., S.P.P., M.L., Q.L., X.X., R.C., P.S., L.Z., Z.E., B.C., D.K. and M.F. performed the experiments. N.M.M., A.K.N.C, L.Y., Y.W., P.P., R.C.R. and C.-W.C. analyzed the data. N.M.M. and A.K.N.C. performed in silico molecular docking. E.M., N.M. and N.V. performed and analyzed the molecular dynamics simulations. S.T.R., J.C., M.A.L., J.S., V.N., R.C.R., M.F. and C.-W.C. provided conceptual input. N.M.M. and C.-W.C wrote the paper. C.-W.C. conceived and supervised the study.

## Competing interests
J.C. is a scientific advisory board member of *Race Oncology*. The other authors declare no competing interests.

## Additional information
**Extended data** is available for this paper at https://doi.org/10.1038/s41594-024-01211-y.

**Correspondence and requests for materials** should be addressed to Chun-Wei Chen.

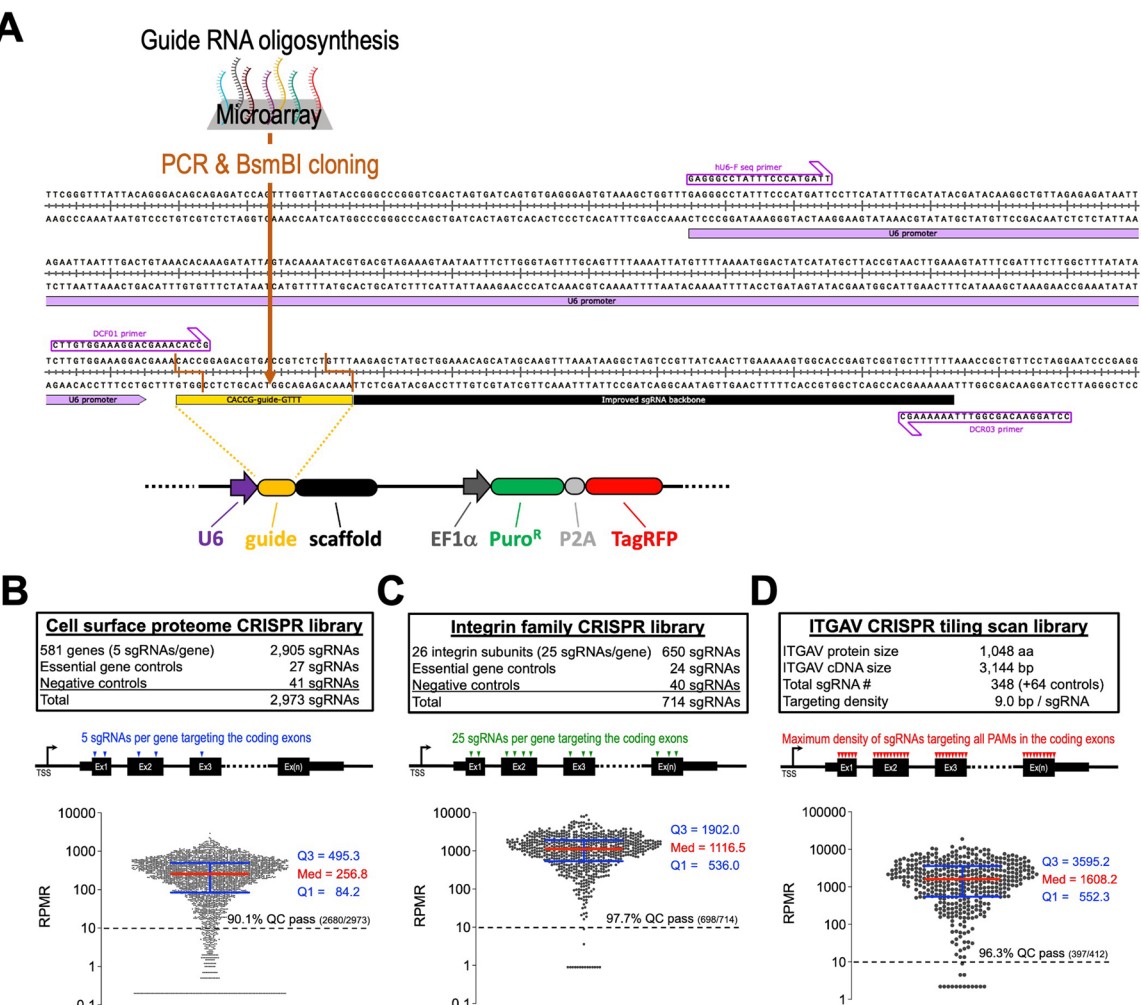

**Extended Data Fig. 1 | CRISPR genetic screen libraries used in this study.**
(**a**) Map of the ipUSEPR vector expressing a sgRNA together with a puromycin-resistant gene (PuroR) and a TagRFP fluorescent protein. Primers for Sanger (hU6-F_seq) and Illumina (DCF01 and DCR03) sequencing are listed. (**b**–**d**) Design and distribution of individual sgRNA frequencies RPMR (reads per million reads) in the CRISPR libraries targeting (B) cell surface proteome genes (n = 2,973 sgRNAs), (C) integrin family genes (n = 714 sgRNAs), and (D) coding regions of ITGAV (n = 412 sgRNAs). (B) 90.1%, (C) 97.7%, and (D) 96.3% of sgRNA in these libraries passed the QC by exhibiting RPMR ≥ 10. Data are represented as median ± interquartile range.

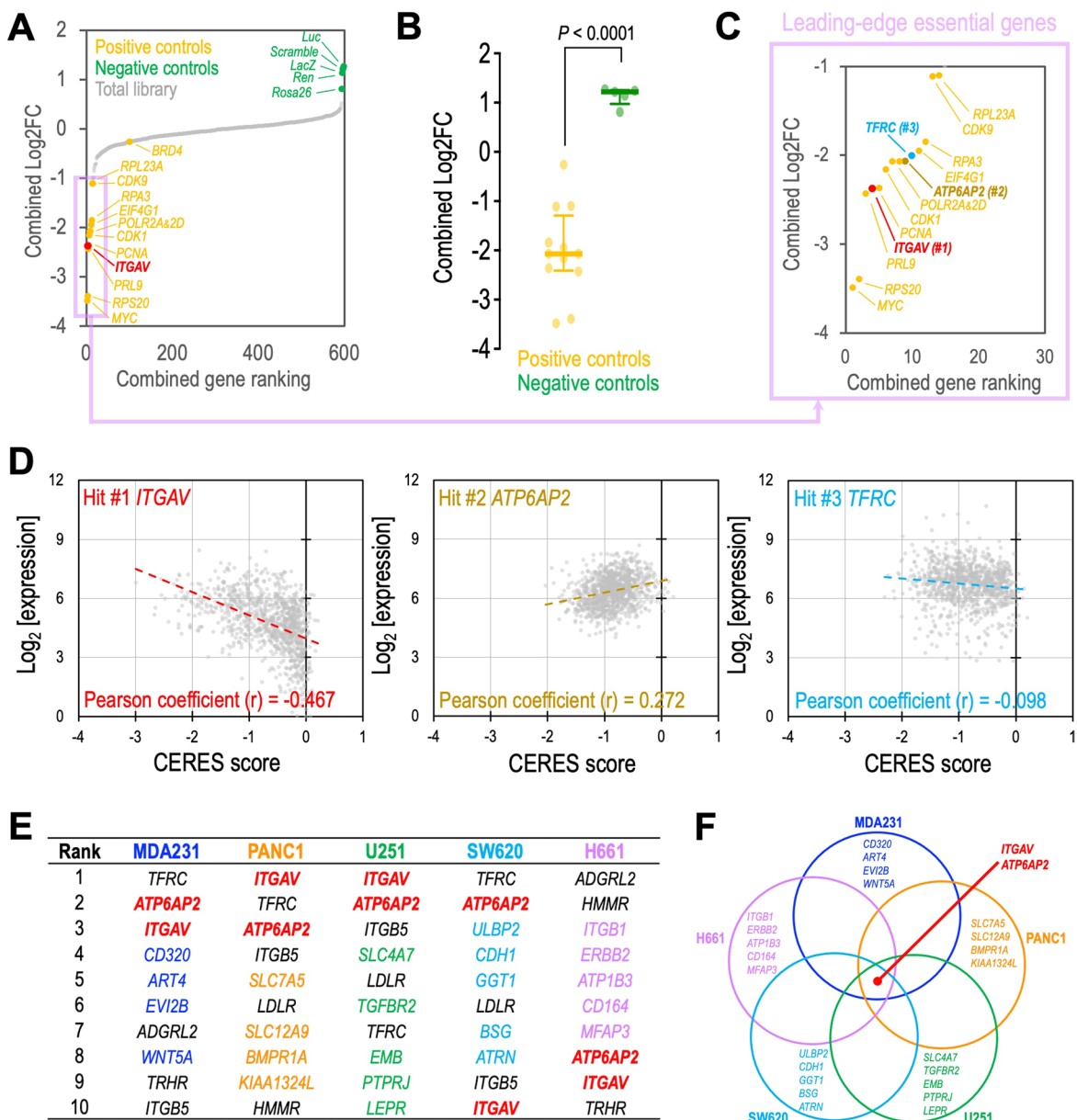

**Extended Data Fig. 2 | Analyses of the surface proteome CRISPR library screens.** (**a**) Combined gene ranking of the cell surface proteome CRISPR screens was calculated by the MAGeCK algorithm. The ranking of ITGAV (red), positive controls (yellow; target common essential genes), negative controls (green; target non-essential sequences), and total library (grey) are indicated. The pink box highlights the leading-edge essential genes with a combined Log2FC below -1.0. (**b**) Distribution of the positive (n = 12 genes) and negative (n = 5 genes) controls in the screen. Data are represented as median ± interquartile range. P value was calculated by two-sided Student's t-test. (**c**) Three surface protein genes (ITGAV, ATP6AP2, and TFRC) were identified as the leading-edge essential genes. (**d**) Correlation of the CERES scores (computational method to estimate gene-dependency levels from CRISPR-Cas9 essentiality screens) and gene expression of ITGAV (left panel), ATP6AP2 (middle panel), and TFRC (right panel) (source: https://depmap.org/portal/; BROAD Institute). The cancer cell dependency on ITGAV is correlated with its expression. (**e**) Top ten candidate hits and (**f**) an overlap plot of the surface proteome CRISPR screens in five cell models. Red (ITGAV and ATP6AP2) indicates the common essential surface proteins in all screened cell types. Other colors (blue, orange, green, cyan, and pink) highlight the cell type-specific candidate genes.

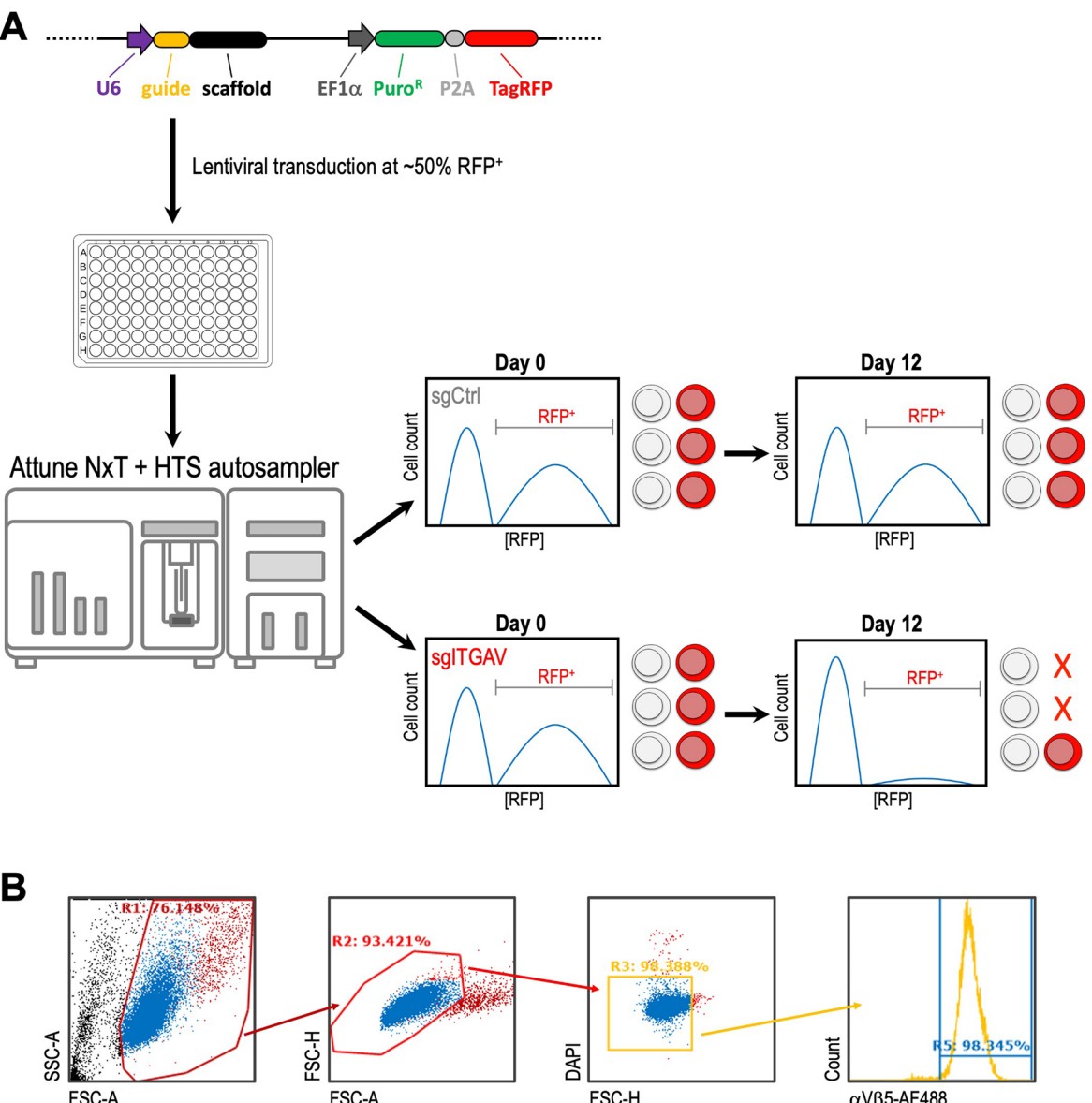

**Extended Data Fig. 3 | Schematic outline of the flow cytometric analysis.** (**a**) RFP growth competition assay (used in Figs. 1e, 2d, 3e): The ipUSEPR vector expresses a sgRNA together with a puromycin-resistant gene (PuroR) and a TagRFP fluorescent protein. The RFP fluorescent signal of live (DAPI⁻) singlet cells was detected by an Attune NxT flow cytometer with an HTS autosampler. The sgRNA targeting a functionally important gene will result in a reduced RFP⁺ population in the culture. (**b**) Gating strategy for detecting the cell surface αVβ5 expression (used in Fig. 5e, f).

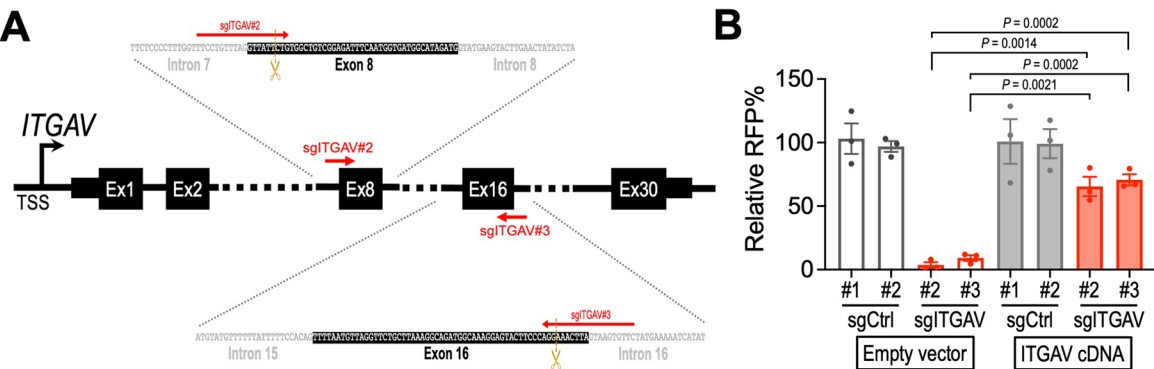

**Extended Data Fig. 4 | The effect of sgITGAV can be reversed by the exogenous ITGAV cDNA.** (**a**) Schematic outline of the ITGAV gene coding region. The recognition sites of sgITGAV#2 and sgITGAV#3 span across the exon-intron junctions. These sgITGAVs only target the endogenous ITGAV coding sequence (with introns) but cannot recognize the ITGAV cDNA sequence (w/o introns), thus allowing the reconstitution of ITGAV through cDNA transduction. (**b**) Transduction of exogenous ITGAV cDNA in MDA231 cells significantly reversed the impact of sgITGAV on cell survival (n = 3 for each group). Data are represented as mean ± s.e.m. P values were calculated by two-sided Student's t-test.

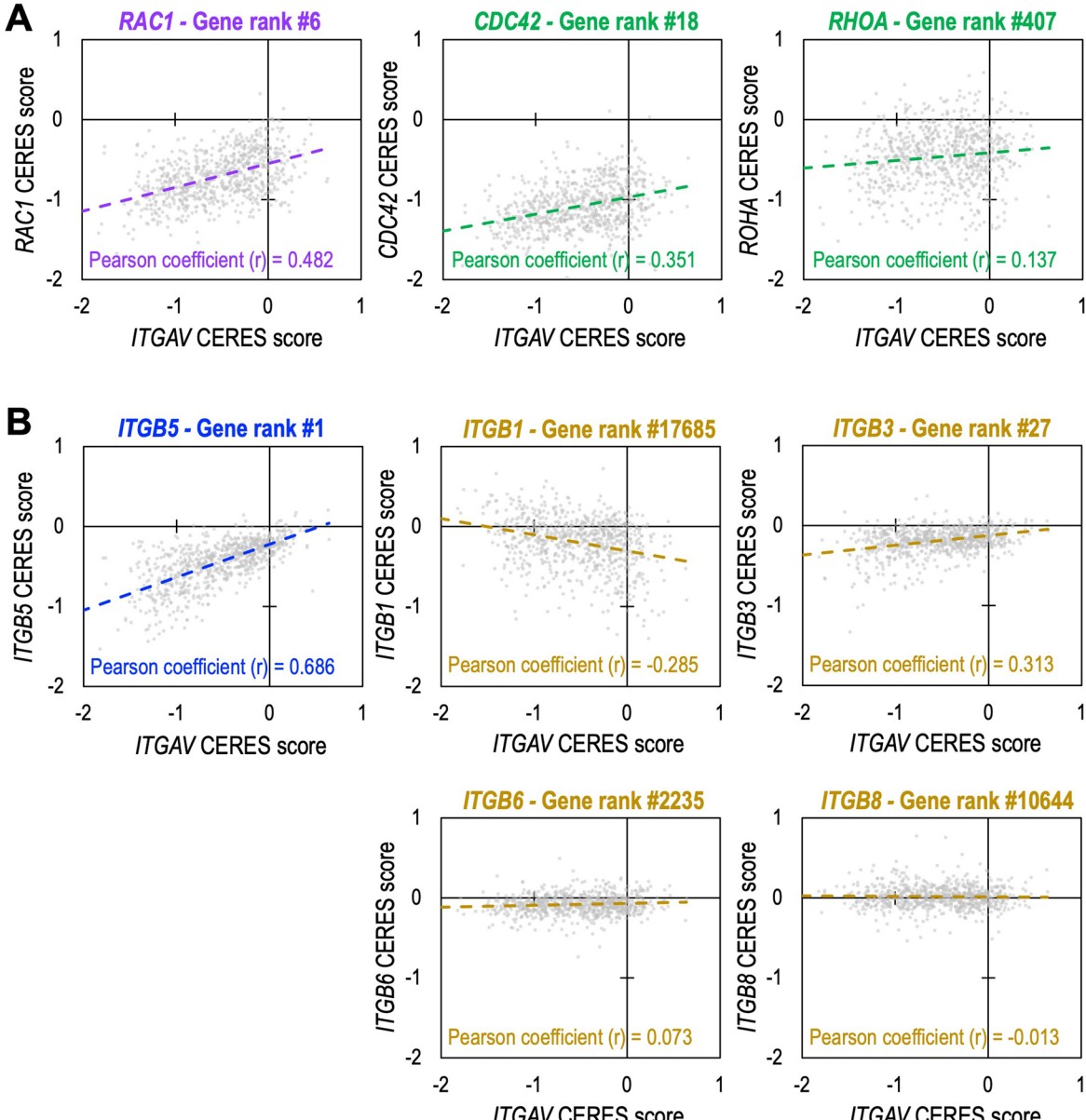

**Extended Data Fig. 5 | Correlation of the CERES scores between ITGAV and other genes.** CERES score is a computational method to estimate gene-dependency levels from CRISPR-Cas9 essentiality screens. The CERES scores of ITGAV (x-axis) and (**a**) Rho small GTPase genes RAC1, CDC42, and RHOA (y-axes; left, middle, and right respectively) and (**b**) ITGB1/3/5/6/8 (y-axes; top-left, top-middle, top-right, bottom-middle, bottom-right, respectively) in 769 cell models (dots) were obtained from the DepMap CRISPR screen consortium database (source: https://depmap.org/portal/; BROAD Institute). A higher Pearson coefficient (*r*) of the CERES scores between two genes indicates a higher likelihood the two genes are co-regulated in the tested cell models. The gene rank number is based on the Pearson coefficient (*r*) of the CERES scores between ITGAV and a total of 17,709 genes tested in the genome-wide CRISPR library screens.

**A**

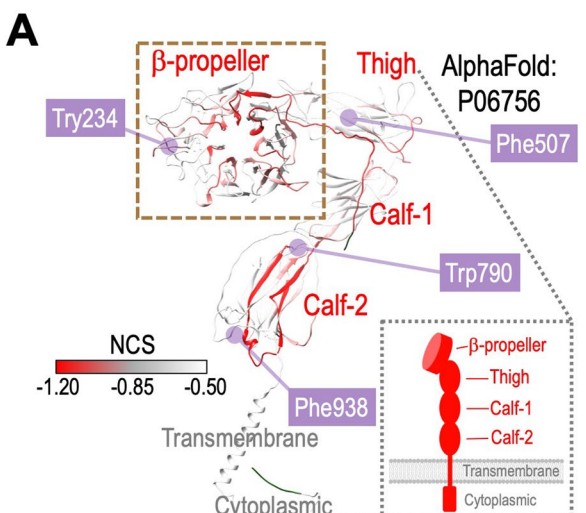

**B**

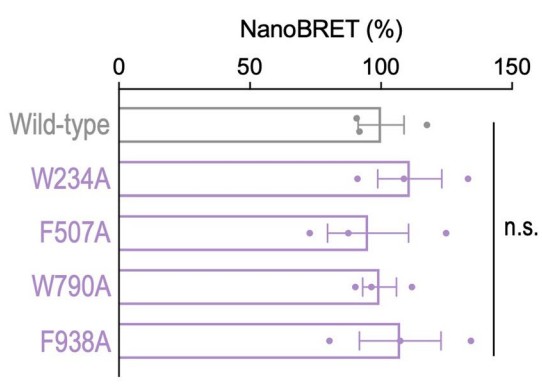

**Extended Data Fig. 6 | Effect of alanine substitution of non-HIP aromatic residues on the NanoBRET assay.** (**a**) The location of four aromatic residues (W234, F507, W790, and F938) outside of HIP were highlighted. (**b**) Alanine substitution of these non-HIP residues (purple; n = 3 for each group) exhibits minimal impact on the NanoBRET signal compared to the wild-type ITGAV (gray; n = 3). Data are represented as mean ± s.e.m. P values were calculated by two-sided Student's t-test.

| Cpd Name | NCI/DTP NSC ID | ΔG° (kJ/mol)* |
|----------|----------------|---------------|
| Cpd_AV2 | 268394 | -15.0 |
| Cpd_AV5 | 613575 | -14.6 |
| Cpd_AV82 | 641239 | -12.8 |
| Cpd_AV84 | 18334 | -12.8 |
| Cpd_AV259 | 94514 | -12.1 |
| Cpd_AV343 | 342443 | -12.0 |
| Cpd_AV377 | 328426 | -11.9 |
| Cpd_AV388 | 59276 | -11.8 |
| Cpd_AV469 | 374119 | -11.6 |

*Predicted by AutoDock Vina

Cpd_AV2 ($C_{32}H_{40}N_4$)
Mwt. 480.7 g/mol
NCI/DTP NSC # 268394

Cpd_AV5 ($C_{35}H_{29}N_7O_5$)
Mwt. 627.7 g/mol
NCI/DTP NSC # 613575

Cpd_AV82 ($C_{29}H_{17}ClF_3NO_5$)
Mwt. 551.9 g/mol
NCI/DTP NSC # 641239

Cpd_AV84 ($C_{42}H_{53}NO_{16}$)
Mwt. 827.9 g/mol
NCI/DTP NSC # 18334

Cpd_AV259 ($C_{36}H_{42}N_6O_2$)
Mwt. 590.8 g/mol
NCI/DTP NSC # 94514

Cpd_AV343 ($C_{38}H_{50}O_{16}$)
Mwt. 762.8 g/mol
NCI/DTP NSC # 342443

Cpd_AV377 ($C_{40}H_{52}O_{17}$)
Mwt. 804.8 g/mol
NCI/DTP NSC # 328426

Cpd_AV388 ($C_{30}H_{45}NO_3$)
Mwt. 467.7 g/mol
NCI/DTP NSC # 59276

Cpd_AV469 ($C_{30}H_{42}O_8$)
Mwt. 530.7 g/mol
NCI/DTP NSC # 374119

**Extended Data Fig. 7 | Information of the candidate ITGAV targeting compounds.** The NCI/DTP NSC identifier, the predicted binding free energy (ΔG°) to ITGAV's β-propeller HIP, and the chemical structure of the top 9 candidate compounds are indicated. The identity (up-right; within one ppm of theoretical value) of Cpd_AV2 was validated by an Orbitrap Fusion Tribrid Mass Spectrometer (Thermo Scientific) at the City of Hope Integrated Mass Spectrometry Shared Resource.

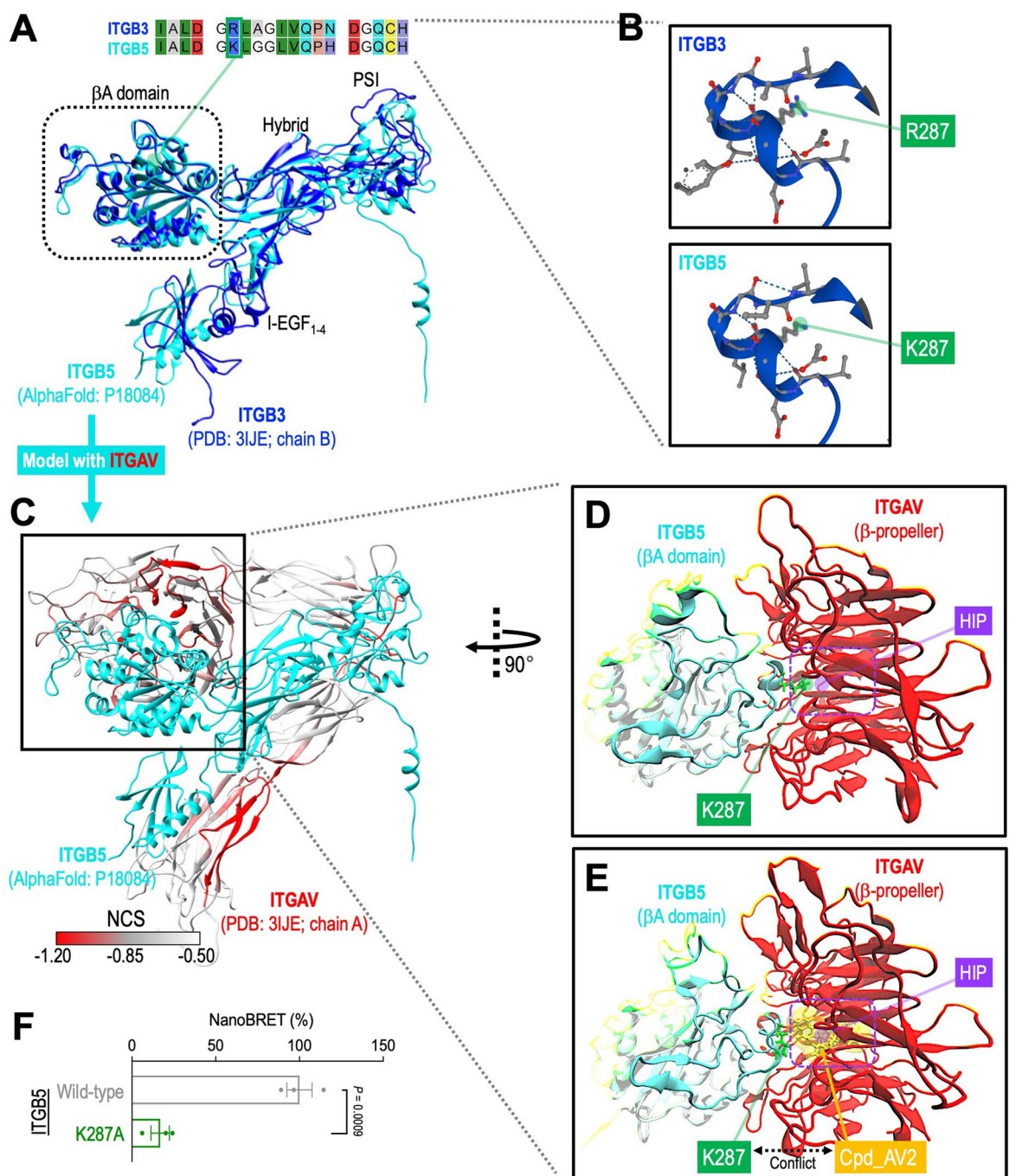

**Extended Data Fig. 8 | Modeling of ITGAV/ITGB5 interaction.** (a) 3D structure of the extracellular domain of ITGB5 was modeled by AlphaFold2 (cyan) and overlaid with the ITGB3 portion of integrin αVβ3 structure resolved by Xiong et al. (PDB ID: 3IJE, chain B; blue). Overall, we observed high concordance of the 3D structures between ITGB3 and ITGB5, including the highly conserved basic amino acid (ITGB3's R287 or ITGB5's K287) in the loop motif of the βA domain highlighted in (b). (c) Modeling of ITGAV/ITGB5 interaction using the AlphaFold2 predicted ITGB5 structure (cyan) and the ITGAV portion of integrin αVβ3 structure (PDB ID: 3IJE, chain A; red). (d and e) Molecular dynamics simulation using GROMACS 2022 with CHARMM36m force field indicates (d) a close contact between ITGB5's K287 and ITGAV's β-propeller HIP pocket (purple box), and (e) the occupancy of Cpd_AV2 (yellow) into ITGAV's HIP pocket disengaged the side chain of ITGB5's K287 from stably interacting with ITGAV. (f) Substitution of ITGB5's K287 with an alanine (K287A) significantly attenuated the ITGAV/ITGB5 NanoBRET signal, highlighting an essential role of this basic residue in integrin αVβ5 assembly (n = 3 for each group). Data are represented as mean ± s.e.m. P value was calculated by two-sided Student's t-test.

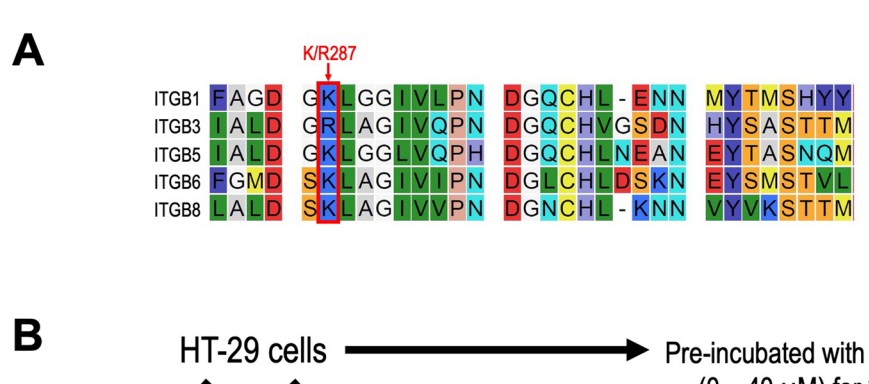

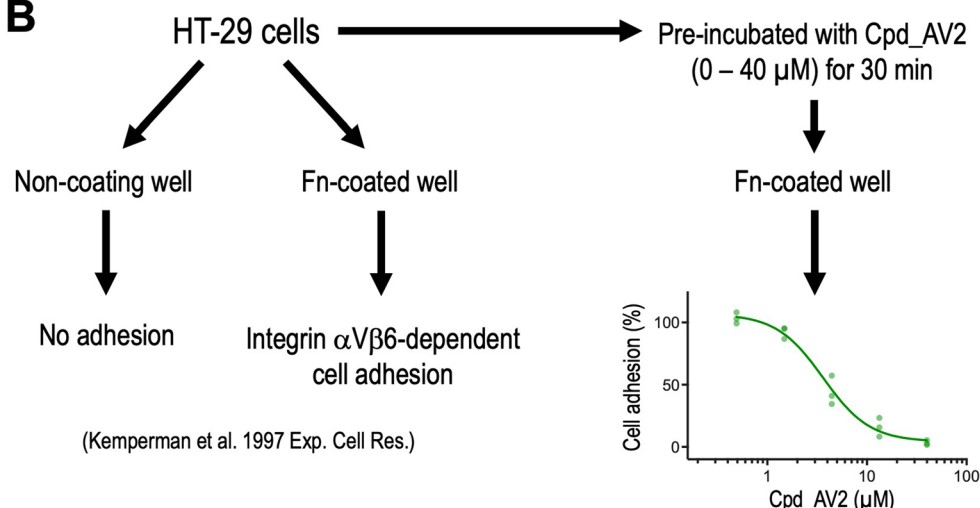

**Extended Data Fig. 9 | Potential impact of Cpd_AV2 on additional ITGAV integrin pairs.** (a) Sequence alignment of ITGAV heterodimer partners ITGB1/3/5/6/8 at the loop motif of their βA domain. The highly conserved basic amino acid (K/R287) encapsulated in ITGAV's β-propeller is labeled. (b) Cpd_AV2 treatment attenuates the integrin αVβ6-mediated adhesion to fibronectin (Fn) in HT-29 colorectal carcinoma cells (n = 3 for each group).

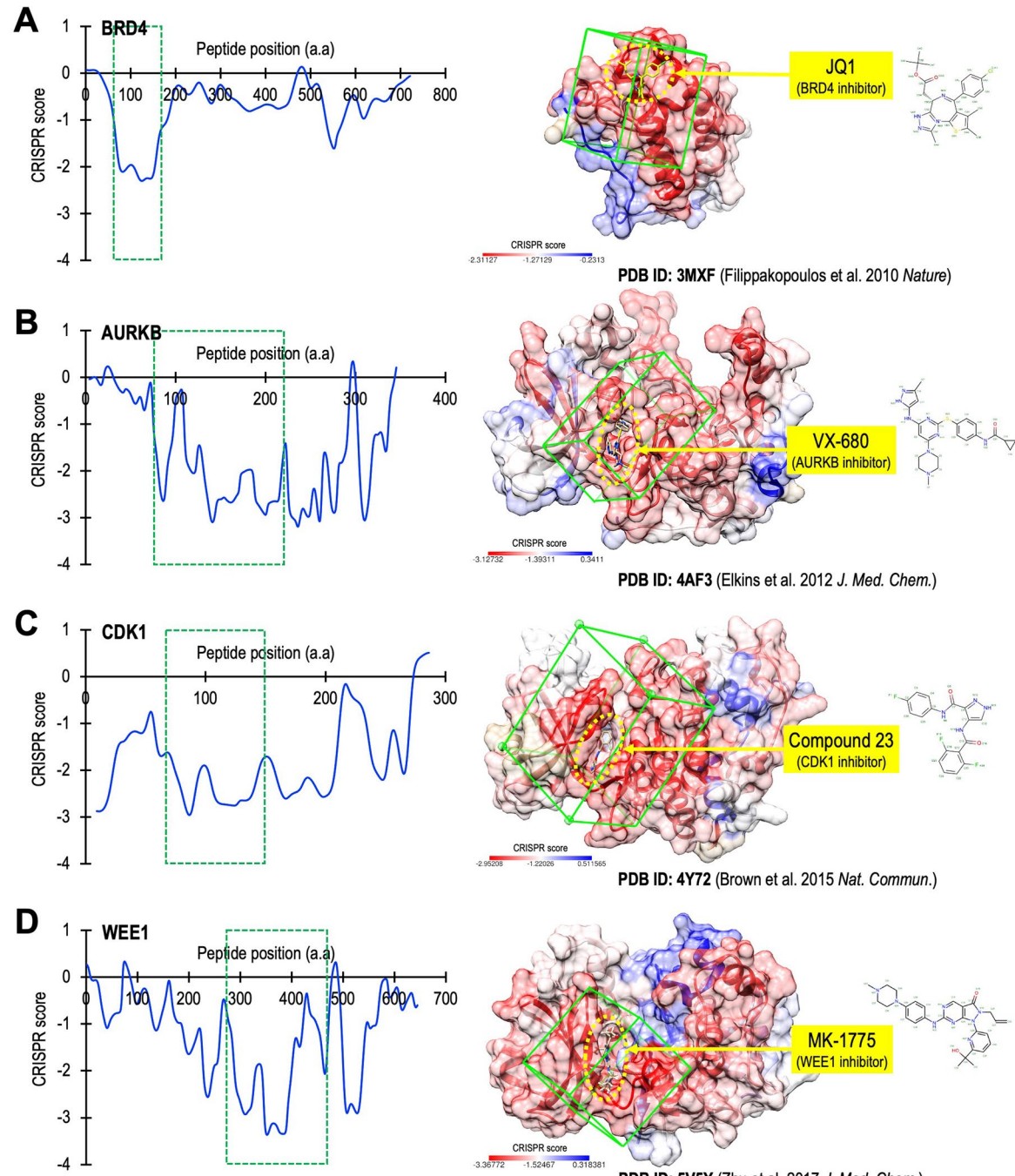

**Extended Data Fig. 10 | CRISPR-TICA evaluation of well-defined drug-targeting pockets.** The smoothened CRISPR tiling data (left panel; blue lines) of (**a**) BRD4, (**b**) AURKB, (**c**) CDK1, and (**d**) WEE1 were obtained from Munoz et al. On the right panels, the green boxes highlight the CRISPR-TICA region of interest based on the CRISPR sensitivity. The yellow arrows indicate the previously reported inhibitors for these proteins.

# nature research

# Reporting Summary

Nature Research wishes to improve the reproducibility of the work that we publish. This form provides structure for consistency and transparency in reporting. For further information on Nature Research policies, see our Editorial Policies and the Editorial Policy Checklist.

## Statistics

For all statistical analyses, confirm that the following items are present in the figure legend, table legend, main text, or Methods section.

| n/a | Confirmed | |
|---|---|---|
| ☐ | ☒ | The exact sample size (*n*) for each experimental group/condition, given as a discrete number and unit of measurement |
| ☐ | ☒ | A statement on whether measurements were taken from distinct samples or whether the same sample was measured repeatedly |
| ☐ | ☒ | The statistical test(s) used AND whether they are one- or two-sided <br> *Only common tests should be described solely by name; describe more complex techniques in the Methods section.* |
| ☒ | ☐ | A description of all covariates tested |
| ☒ | ☐ | A description of any assumptions or corrections, such as tests of normality and adjustment for multiple comparisons |
| ☐ | ☒ | A full description of the statistical parameters including central tendency (e.g. means) or other basic estimates (e.g. regression coefficient) AND variation (e.g. standard deviation) or associated estimates of uncertainty (e.g. confidence intervals) |
| ☐ | ☒ | For null hypothesis testing, the test statistic (e.g. *F*, *t*, *r*) with confidence intervals, effect sizes, degrees of freedom and *P* value noted <br> *Give P values as exact values whenever suitable.* |
| ☒ | ☐ | For Bayesian analysis, information on the choice of priors and Markov chain Monte Carlo settings |
| ☒ | ☐ | For hierarchical and complex designs, identification of the appropriate level for tests and full reporting of outcomes |
| ☐ | ☒ | Estimates of effect sizes (e.g. Cohen's *d*, Pearson's *r*), indicating how they were calculated |

*Our web collection on statistics for biologists contains articles on many of the points above.*

## Software and code

Policy information about availability of computer code

| Data collection | Sequencing data was collected on NextSeq 550 and NovaSeq 6000 (Illumina) |
|---|---|
| Data analysis | The computational codes/tool packages used in this study include Genetic Perturbation Platform (BROAD Institute), Bowtie2 (Johns Hopkins University), UCSF Chimera 1.15 (UC San Francisco), Attune NxT v3.1.2 (ThermoFisher), GSEA v4.1.0 (UC San Diego and BROAD Institute), FASTQC v0.11.8, Burrows-Wheeler Aligner v0.7.17, MACS2 v2.1.1, Samtools v1.10, STAR v2.6.1d, featureCounts v1.6.4, edgeR v4.0.1, deepTools v3.5.1, IGV 2.14.0 (BROAD Institute), PyMOL v2.0.4 (Schrödinger, LLC), PDB2PQR server, AutoDockTools, AutoSite, Open Babel v2.4.1, PyRx v0.9.7, AutoDock Vina v1.1.2, Raccoon2, JTSA (https://paulsbond.co.uk/jtsa), Bio-Rad ChemiDoc MP (Bio-Rad), MAGeCK v0.5.9.2. IC50 and two-sided Student's t-test were performed using Prism 9 (GraphPad). |

For manuscripts utilizing custom algorithms or software that are central to the research but not yet described in published literature, software must be made available to editors and reviewers. We strongly encourage code deposition in a community repository (e.g. GitHub). See the Nature Research guidelines for submitting code & software for further information.

## Data

Policy information about availability of data

All manuscripts must include a data availability statement. This statement should provide the following information, where applicable:
- Accession codes, unique identifiers, or web links for publicly available datasets
- A list of figures that have associated raw data
- A description of any restrictions on data availability

The RNA-seq data generated in this study are available via Gene Expression Omnibus (GEO) under accession GSE231339. All the data supporting the findings of this study are included in this article and its Supplementary Information. Three-dimensional protein structure (PDB ID: 3IJE) was obtained from the Research Collaboratory for Structural Bioinformatics Protein Data Bank (RCSB PDB; https://www.rcsb.org). ITGAV expression data in breast, pancreas, brain, colon, and lung

cancers was obtained from Gene Expression database of Normal and Tumor tissues (GENT2 database: http://gent2.appex.kr/gent2/). Additional data that support the findings of this study are provided in the Supplementary Information.

# Field-specific reporting

Please select the one below that is the best fit for your research. If you are not sure, read the appropriate sections before making your selection.

☒ Life sciences  ☐ Behavioural & social sciences  ☐ Ecological, evolutionary & environmental sciences

For a reference copy of the document with all sections, see nature.com/documents/nr-reporting-summary-flat.pdf

# Life sciences study design

All studies must disclose on these points even when the disclosure is negative.

| | |
|---|---|
| Sample size | Fig. 1A-C: total 2973 sgRNAs library screen performed in 5 cell line models. n = 3 was chosen to allow two-sided Student's t-test.<br>Fig. 1D: 2 independent sgCtrl sequences and 3 independent sgITGAV sequences. Sample size was chosen based on the available independent sgRNA numbers.<br>Fig. 1E: 2 independent sgCtrl sequences and 3 independent sgITGAV sequences. n = 3 for each sgRNA group was chosen to allow two-sided Student's t-test.<br>Fig. 1F,G: n = 3 for each sgRNA group was chosen to allow two-sided Student's t-test.<br>Fig. 1H: n = 927 patients for each ITGAV(high) and ITGAV(low) groups. Sample size was chosen based on the total available data from the GEPIA database.<br>Fig. 2A: 17110 genes tested in 769 cell models. Sample size was chosen based on the total available data from the DepMap database.<br>Fig. 2B: n = 3 for each sgRNA group was chosen to allow GSEA analysis.<br>Fig. 2C: 2 independent sgCtrl sequences and 3 independent sgRAC1 sequences. Sample size was chosen based on the available independent sgRNA numbers.<br>Fig. 2D: 2 independent sgCtrl sequences and 3 independent sgRAC1 sequences. n = 3 for each sgRNA group was chosen to allow two-sided Student's t-test.<br>Fig. 2E,F: n = 3 for each sgRNA group was chosen to allow two-sided Student's t-test.<br>Fig. 2H: sgCtrl (n = 30), sgITGAV (n = 42), and sgRAC1 (n = 33). Sample size was chosen based on available cell number for each sgRNA group.<br>Fig. 3B-D: total 712 sgRNAs library screen performed in 2 cell line models. n = 3 was chosen to allow two-sided Student's t-test.<br>Fig. 3E: 2 independent sgCtrl sequences and 3 independent sgRNA sequences for each ITGB gene. n = 3 for each sgRNA group was chosen to allow two-sided Student's t-test.<br>Fig. 3F: 2 independent sgCtrl sequences and 3 independent sgITGB5 sequences. Sample size was chosen based on the available independent sgRNA numbers.<br>Fig. 3G,H: n = 3 for each sgRNA group was chosen to allow two-sided Student's t-test.<br>Fig. 3I: 17110 genes tested in 769 cell models. Sample size was chosen based on the total available data from the DepMap database.<br>Fig. 4A,B: total 412 sgRNAs library screen performed in MDA231-Cas9 cells. n = 3 was chosen to allow two-sided Student's t-test.<br>Fig. 4F: n = 3 for each ITGAV cDNA group was chosen to allow two-sided Student's t-test.<br>Fig. 5B: total 128562 compounds. Sample size was chosen based on the total available compounds collected by the NCI/DTP Open Chemicals Repository.<br>Fig. 5C: total 500 compounds with the top binding energy were chosen for CellTiterGlo and CCK8 assays. n = 3 for each condition was chosen to allow two-sided Student's t-test.<br>Fig. 5F: n = 4 in this experiment based on the available cell cultures. A minimum of n = 3 is required for IC50 test.<br>Fig. 5G,H: n = 3 for each treatment group was chosen to allow two-sided Student's t-test.<br>Fig. 5J: Control (n = 30) and Cpd_AV2 (n = 33). Sample size was chosen based on available cell number for each treatment group.<br>Fig. 6A: Representative gel of 2 independent protein purification experiments.<br>Fig. 6B: Vehicle (724 data points) and Cpd_AV2 (724 data points) was chosen based on the available data from 25°C to 80°C.<br>Fig. 6D: n = 3 for each treatment group was chosen to allow two-sided Student's t-test.<br>Fig. 6E: n = 3 for each treatment group was chosen to allow two-sided Student's t-test and IC50 test.<br>Fig. 6F: n = 3 for each treatment group was chosen to allow IC50 test. |
| Data exclusions | No data point was excluded. |
| Replication | For Fig. 1A-C, 3B,C and 4A,B, triplicated library screens were performed. For other experiments, a minimum of 2 independent experiments were performed and all attempts at replication were successful. |
| Randomization | In cell culture experiment, an initial cell culture was split into individual cultures randomly with an equal seeding density. Each culture received a sgRNA, cDNA, CRISPR library, or compounds without predetermination. |
| Blinding | The same group of researchers designed, operated, and analyzed the experiments. Therefore, they were aware of the treatment conditions while executing and analyzing the experiments. |

# Reporting for specific materials, systems and methods

We require information from authors about some types of materials, experimental systems and methods used in many studies. Here, indicate whether each material, system or method listed is relevant to your study. If you are not sure if a list item applies to your research, read the appropriate section before selecting a response.

## Materials & experimental systems

| n/a | Involved in the study |
|---|---|
| ☐ | ☒ Antibodies |
| ☐ | ☒ Eukaryotic cell lines |
| ☒ | ☐ Palaeontology and archaeology |
| ☒ | ☐ Animals and other organisms |
| ☒ | ☐ Human research participants |
| ☒ | ☐ Clinical data |
| ☒ | ☐ Dual use research of concern |

## Methods

| n/a | Involved in the study |
|---|---|
| ☒ | ☐ ChIP-seq |
| ☐ | ☒ Flow cytometry |
| ☒ | ☐ MRI-based neuroimaging |

# Antibodies

**Antibodies used**

Western blot: primary antibodies against ITGAV (4711, Cell Signaling Technology, 1:1000), ITGB5 (3629, Cell Signaling Technology; 1:1000), RAC1 (4651, Cell Signaling Technology; 1:1000), and beta-actin (ab8226, Abcam; 1:5000) at 4°C overnight. After washing, the membranes were incubated with HRP-conjugated goat anti-mouse (31430, Invitrogen; 1:10,000) or goat anti-rabbit (31460, Invitrogen; 1:10,000) IgG antibodies at room temperature for 1 hour. The chemiluminescent signals were detected using a ChemiDoc imaging system (Bio-Rad). The cell surface integrin αVβ5 was recognized by a mouse monoclonal anti-human αVβ5 antibody (clone P1F76; sc-13588, Santa Cruz Biotech; 1:200) and stained by AF488-conjugated donkey anti-mouse IgG (ab150105, Abcam) secondary antibody.

**Validation**

1. The specificity of anti-human-ITGAV (4711, Cell Signaling Technology, 1:1000), anti-human-ITGB5 (3629, Cell Signaling Technology; 1:1000), anti-human-RAC1 (4651, Cell Signaling Technology; 1:1000) to their target proteins was confirmed by Western blots with CRISPR depletion of the targeted protein in human cells as shown in Fig. 1D, 2C, 3F, and Suppl. Fig. 4.
2. The specificity of anti-human beta-actin (ab8226, Abcam; 1:5000) was evaluated by Western blot of human cell samples and observed a single band at the expected molecular weight (~40 kDa) as shown in Suppl. Fig. 4.
3. HRP-conjugated goat anti-mouse (31430, Invitrogen; 1:10,000) and goat anti-rabbit (31460, Invitrogen; 1:10,000) IgG secondary antibodies have been used in our lab for multiple projects, including the detection of protein IP and co-IP. These secondary antibodies have shown high specificity to only detect the primary antibodies from their targeted species (mouse, rabbit).
4. The specificity of monoclonal anti-human αVβ5 antibody (clone P1F76; sc-13588, Santa Cruz Biotech; 1:200) to cell surface integrin αVβ5 (stained by AF488-conjugated donkey anti-mouse IgG; ab150105, Abcam) was confirmed by flow cytometry of human cells with sgCtrl vs. sgITGAV as shown in Fig. 5E. Another mouse monoclonal anti-human αVβ5 antibody (clone P1F6; 920005, Biolegend; AF647-conjugated) was used to validate the αVβ5 flow cytometry results.

# Eukaryotic cell lines

Policy information about cell lines

**Cell line source(s)**

HEK293, PANC1, and SW620 cells were obtained from the American Type Culture Collection (ATCC). MDA231 (i.e., MDA-MB-231) cells were obtained from Dr. Mingye Feng (City of Hope Cancer Center; original commercial source: ATCC). H661 cells were obtained from Dr. Jun Qi (Dana Farber Cancer Institute; original commercial source: ATCC). U251 cells were obtained from Dr. Mike Chen (City of Hope Cancer Center; original commercial source: European Collection of Authenticated Cell Cultures [ECACC]).

**Authentication**

1. HEK293, PANC1, and SW620 cells were directly purchased from ATCC. As a biological resource center, ATCC comprehensively performs authentication and quality-control tests on all distribution lots of cell lines using short tandem repeat (STR) profiling.
2. MDA231, H661, U251 cells were obtained from our collaborators and were not further authenticated.

**Mycoplasma contamination**

Plasmocin was added in all culture medium to prevent mycoplasma contamiation. All cell lines tested negative for mycoplasma contamination using a Mycoplasma PCR Detection Kit (Abm cat# G238).

**Commonly misidentified lines**
(See ICLAC register)

No commonly misidentified cell lines were used in this study.

# Flow Cytometry

## Plots

Confirm that:

☒ The axis labels state the marker and fluorochrome used (e.g. CD4-FITC).

☒ The axis scales are clearly visible. Include numbers along axes only for bottom left plot of group (a 'group' is an analysis of identical markers).

☒ All plots are contour plots with outliers or pseudocolor plots.

☒ A numerical value for number of cells or percentage (with statistics) is provided.

## Methodology

**Sample preparation**

For competition cell culture assays, Cas9-expressing cells were transduced with the ipUSEPR (RFP+) sgRNA constructs in 96-well plates at ~50% infection. Relative RFP% refers to percentages of RFP+ cells over time after lentiviral infection, which was normalized to the RFP+% on day 0 (i.e., 48 hours after the lentiviral infection). The cell cycle was measured by Click-iT Plus EdU Alexa Fluor 647 Assay Kits (C10634, Invitrogen). Cells were exposed to 10 μM EdU at 37°C for 2 hours, and the percentage of cells in the S phase was defined by EdU-positive cells over the total singlet cells. Cellular apoptosis was detected using Annexin V Apoptosis Detection Kit (50-112-9048, Invitrogen). Live cells were defined by 4',6-diamidino-2-phenylindole (DAPI; D1306, Invitrogen) dye exclusion. The cell surface integrin αVβ5 was detected by a mouse monoclonal anti-human αVβ5 antibody (sc-13588, Santa Cruz Biotech; 1:200).

**Instrument**

Attune NxT flow cytometer with autosampler (ThermoFisher).

**Software**

Attune NxT v3.1.2 (ThermoFisher).

**Cell population abundance**

The RFP, αVβ5 properties were measured over the live/singlet cell population.

**Gating strategy**

FSC/SSC was used to get actual cells. FSC-A/FSC-H was used to get singlet. FSC/DAPI was used to gate live cells. Non-stained (or non-transduced) cells were used as negative controls for gating.

☒ Tick this box to confirm that a figure exemplifying the gating strategy is provided in the Supplementary Information.

