## [Peer Review File · Nature Structural & Molecular Biology]

Peer Review Information

Manuscript Title: A novel class of inhibitors that disrupts the stability of integrin heterodimers identified by CRISPR-tiling instructed genetic screens

Corresponding author name(s): Chun-Wei Chen

Reviewer Comments & Decisions:

Decision Letter, initial version:

Message: 21st Jun 2023

Dear Dr. Chen,

Thank you again for submitting your manuscript "CRISPR-tiling instructed structural genetic screen identifies a novel class inhibitor that disrupts the integrin heterodimer stability". I apologise for the delay in responding, which resulted from the difficulty in obtaining suitable referee reports. Nevertheless, we now have comments (below) from the 2 reviewers who evaluated your paper. In light of these reports, we remain interested in your study and would like to see your response to the comments of the referees, in the form of a revised manuscript.

You will see that, while both experts appreciate the robustness and elegance of the CRISPR screens and the CRISPR-TICA method, some important concerns are raised, which need to be addressed with major revisions. Firstly, Reviewer #1 (R#1, points 2 and 4), provides relevant guidance on how to further boost the impact of the gene tiling/CRISPR screen presented. Additionally, both reviewers raise important issues about the validation part that must be addressed, including the absence of crucial controls with pertinent mechanistic ramifications. Reviewer #2 notes the absence of showing that Cpd-AV2 induces loss of dimerization in the NanoBRET assay (or elsewhere) and R#1 (point 5) points out that other import controls are missing. Convincingly demonstrating that the investigated compounds that lead to apoptosis indeed disrupt dimerisation will be of critical importance. Furthermore, R#2 requests that a more widely acceptable and better benchmarked antibody is used to assess avb5 expression following treatment. We editorially agree that addressing these and other technical issues will be paramount for a successful peer-review process of this work. Importantly, and we are also of the same opinion that this would significantly elevate the value of the manuscript, R#1 requests that rescue experiments are performed in the KO background to validate its essentiality in cancer cells (R#1 point 3). Finally, R#1 proposes that an AF2 prediction of the structure of the interaction interface would better illustrate the value of the functional findings and we

also deem that this would boost the work.

As always, please be sure to address/respond to all concerns of the referees in full in a point-by-point response and highlight all changes in the revised manuscript text file.

We appreciate the requested revisions are extensive. We thus expect to see your revised manuscript within 6 months. If you cannot send it within this time, please let us know. We will be happy to consider your revision as long as nothing similar has been accepted for publication at NSMB or published elsewhere. Should your manuscript be substantially delayed without notifying us in advance and your article is eventually published, the received date would be that of the revised, not the original, version.

Reporting Summary:

When submitting the revised version of your manuscript, please pay close attention to our [href="https://www.nature.com/nature-portfolio/editorial-policies/image-integrity">Digital Image Integrity Guidelines. and to the following points below:](https://www.nature.com/nature-portfolio/editorial-policies/image-integrity)

SOURCE DATA: we urge authors to provide, in tabular form, the data underlying the graphical representations used in figures. This is to further increase transparency in data reporting, as detailed in this editorial (<http://www.nature.com/nsmb/journal/v22/n10/full/nsmb.3110.html>). Spreadsheets can be submitted in excel format. Only one (1) file per figure is permitted; thus, for multi-paneled figures, the source data for each panel should be clearly labeled in the Excel file; alternately the data can be provided as multiple, clearly labeled sheets in an Excel file. When submitting files, the title field should indicate which figure the source data pertains to. We encourage our authors to provide source data at the revision stage, so that they

are part of the peer-review process.

We require deposition of coordinates (and, in the case of crystal structures, structure factors) into the Protein Data Bank with the designation of immediate release upon publication (HPUB). Electron microscopy-derived density maps and coordinate data must be deposited in EMDB and released upon publication. Deposition and immediate release of NMR chemical shift assignments are highly encouraged. Deposition of deep sequencing and microarray data is mandatory, and the datasets must be released prior to or upon publication. To avoid delays in publication, dataset accession numbers must be supplied with the final accepted manuscript and appropriate release dates must be indicated at the galley proof stage. Please find the complete NRG policies on data availability at <http://www.nature.com/authors/policies/availability.html>.

[redacted]

Sincerely,

Dimitris Typas
Associate Editor
Nature Structural & Molecular Biology
ORCID: 0000-0002-8737-1319

Referee expertise:

Referee #1: CRISPR gene tiling, cancer genetics

Referee #2: Integrins (mechanisms and drug development)

Reviewers' Comments:

Reviewer #1:

Remarks to the Author:

Mattson et al. conducted a series of CRISPR screens to identify surface proteins essential for cancer cell survival using surface proteome library followed by integrin-focused library, and to explore potential drug target structure using tiling sgRNA library on ITGAV. They revealed a critical role of ITGAV-ITGB5 dimerization for cancer cell expansion and characterized an integral pocket within ITGAV's beta-propeller domain for the dimerization. Further, they combined compound docking algorithm with tiling-sgRNA screens to identify a compound that inhibit the integrin signaling via heterodimer dissociation. In general, the high-throughput screens and computational methods used in this study are well designed and technologically sound. The method (CRISPR-TICA) that combines tiling screen and compound docking for drug discovery is interesting and hold potential for future applications. The manuscript is well-written with a clear storyline. Based on my expertise, the comments below are focused on the design, interpretation, and validation of high-throughput CRISPR screens, as well as the computational methods employed in this study.

Major:

1. The authors claimed that "ITGAV as a vulnerability in multiple cancers". They made the statement based on i) ITGAV is essential for cancer cell viability; ii) high ITGAV expression is associated with poor survival prognosis. These lines of evidence, however, are insufficient to support the argument of "cancer vulnerability". To confirm "cancer vulnerability", the authors should demonstrate: a) ITGAV is over-expressed in at least a subset of patient tumor samples compared to normal cells; b) cancer cell dependency on ITGAV is correlated with its expression (I found this is true in Depmap data).
2. The authors used a cell surface proteome library for the first screen. Since this is an unbiased large-scale screen, I'd expect more information to be provided and discussed on the screen, including: i) the QC outcome of fold-change by positive and negative controls, which is provided by MaGeCK (this can be shown in supplementary figure); ii) Other than ITGAV, what are the other top hits of surface proteome essential for cancer cells? Of note, ITGAV is essential for all cell lines. Are there other proteins essential for a subset of cell lines? These proteins may also be interesting to the readers. A summary of the screen hits is preferred such that the manuscript could provide an overview of the functional impact of surface proteins on cancer cell viability phenotype.
3. Because ITGAV is the key protein in this study, a rescue experiment is necessary to validate its essentiality for cancer cell viability, in addition to the KO experiments shown in Figure 1.
4. To me, the CRISPR-TICA method is a simple combination of tiling-sgRNA screen and docking algorithm, which is not completely novel in methodology. Nevertheless, the study provides a good prototype of utilizing tiling-sgRNA screen for drug discovery. To demonstrate its potential utility for future applications, authors could provide a few more examples in which a tiling screen highlight the known drug-targeting pockets. This can be done by performing similar analysis on published tiling-sgRNA screen data (e.g. PMID 27260157). Such examples will strengthen the impact and generality of the method.
5. For the mutagenesis analysis with NanoBRET (Fig. 4F), negative controls with mutations of aromatic residues outside the pocket are preferred to demonstrate the critical function

of the pocket for dimerization.

6. Ideally, it is preferred to show the crystal structure of ITGAV-ITGB5 interaction via the identified pocket. If this is too laborious for the revision, an alternative way is to use AlphaFold2 to predict the interaction interface.

Minor:

1. It seems to me that the 2nd screen is sort of redundant to the 1st screen because the genes and cell lines were included in the 1st screen. What's the rationale of performing the 2nd screen? Is there any hit identified in the 2nd screen but not in the 1st screen due to the difference of guide per gene?

2. Font format and size are inconsistent. Some are too big, and some are too small. Reformatting is necessary before publication.

Reviewer #2:

Remarks to the Author:

A. Summary of key results

Mattson et al. use an unbiased CRISPR KO screening technology and identify Integrin alphaVbeta5 ("avb5") as a cell surface protein preventing cancer cell line apoptosis. KO of RAC1 (a known avb5 downstream partner) had a similar effects. Elegant CRISPR gene-tilling identified hydrophobic residues surrounding the central pore of the ITGAV chain β -propellor as necessary for avb5 activity. Alanine-substitution of these residues prevented alphaV and beta5 chains from forming heterodimers. Molecular docking of an 1E5 compound library onto this region identified model leads, and these reduced cell viability of MDA-MB-231 cells. One compound, AV2, clearly reduced cell surface expression of alphaVbeta5, and at similar concentration induced apoptosis and cell rounding.

Recombinant alphaV interacted with Cpd-AV2, and AV2 reduced cell surface expression of avb5 and cell viability compared to the avb5/avb3 competitive inhibitor cilengitide. The authors conclude that their method identifies novel integrin inhibitors based on an ability to disrupt alpha-beta chain dimerization.

B. Originality and significance

That avb5 can protect tumor-derived cells from apoptosis is known (e.g. Uhm et al. Clin Cancer Res (1999) 5 (6):1587; Lane et al. Oncogene (2010)29:3519), but the CRISPR unbiased screen confirming this is novel. The original CRISPR mediated search for novel inhibitors of integrin function finding regions on the alpha-v chain critical for dimerization is a novel and promising approach. And the molecular docking screen identifying candidates which trigger apoptosis via these regions sites is elegant and original. Overall, the team attempt to show that this approach in a general unbiased way can identify an so far unexpected targetable region on integrins, and use docking to credibly identify small molecules that can bind this region.

C. Data and methodology

The data is well described and generally well presented. The methodology is certainly original, and the authors show very well once again the power of CRISPR screening approaches. Adequate positive and negative controls are included in the screen. It is the

unbiased nature of their screens that make them interesting. Although the target maybe less so, recalling that at least one drug, the pan-av inhibitor antibody Abituzumab had minimal observable avb5-mediated effect (Elez et al. *Ann Oncol.* (2015);26(1):132).

The approach seems valid, but perhaps the interpretation of the effect less so.

D, Appropriate use of statistics and treatment of uncertainties.

No issues.

E. Conclusions: robustness, validity, reliability

May we start with integrin dimerization? The authors claim that by messing up the hydrophobic rings of the β -propellor they affect dimerization. They have an excellent carefully constructed dimerization FRET-like assay, NanoBRET easily able to detect dimerization, and they show that alanine substituting their carefully discovered CRISPR identified hydrophobic residues (Fig 4F) disrupt dimerization. The cage of hydrophobic β -propellor residues enclosing the Arg261 (avb3)/Lys(b1,b5,b6,b8) residue of the beta-chains and likely holding them in place was noted in our original structure paper. This is a direct demonstration that its disruption affects heterodimerization.

But they do not show that their apoptosis inducing compounds especially Cpd-AV2 disrupts dimerization. Alanine scan- disrupting the selected hydrophobic residues may prevent en-caging of the necessary beta-chain Lys261 residue (the postulated inhibition of dimerization). Or the alphaV chains aren't folding correctly, and no β -propellor exists. It may well be that Cpd-AV2 indeed disrupts dimerization. But it is not shown.

The use of an EColi derived alpha chain is odd. Integrin avb5 is commercially available. As far as I am aware, intact integrin chains have not been shown to fold correctly in bacteria. Given the high degree of glycosylation this is no big surprise. Only mammalian recombinant integrins have demonstrably correct folding and activity. Does the authors' "refolded" chain interact with conformation-dependent anti-av antibodies, for example? The target compounds, like AV2, look pretty hydrophobic, so perhaps it is not surprising that they interact (Fig6A-C). with hydrophobic patches on the refolded E.coli recombinant alpha chain.

In short I have doubts about the validity of the conclusion that the selected reagents affect dimerization.

Integrin avb5 as critical: the cell culture conditions used in the selection assays ensure that a vitronectin substrate is offered to the cells, which all predominantly (4 epithelial derived; one avb5/avb3 glioma) use avb5 to attach to vitronectin. So removing this integrin by KO forces cells to round up, as shown, and sometimes to detach. This would tend to favour apoptosis.

F. Suggested improvements, experiments data for possible revision.

The authors should please show the readers the NanoBRET assay in at least MDA-MB-231 cells exposed to various concentrations of Cpd-AV2 with a time course. This would directly show whether AV2 induces loss of dimerization.

Antibodies. Please could the authors refer to protein concentrations of the antibodies they use, not dilutions? The readers have no idea of the starting concentrations, so the dilutions are sadly meaningless.

I was puzzled that the authors used an obscure antibody for the critical cell-surface FACS data investigating avb5 expression following compound treatment. I would suggest a well characterized antibody (perhaps P1F6?) in this crucial assay would be more definitive?

The Western blots showing gel slices should please be complemented by images of the entire gel blots, so that the readers may assess the specificity of the antibodies used directly. In Fig2H and 5J the authors mention "Cell dimension". I think they may mean projected cell area/diameter. Please could they clarify?

According to the authors hypothesis, the Cpd-AV5 interacts with the alphaV chain. So it is expected that it would disrupt heterodimers of at least avb1, avb6 and avb8 too, they also use Lys encapsulated in the alphaV chain b-propeller. Might a proof of concept involve disrupting an avb6 dependent adhesion (HT29 cells on fibronectin springs to mind)?

G. References.

Appear appropriate

H. Clarity and context.

The work is well written and described, with the exceptions noted.

Simon Goodman

Author Rebuttal to Initial comments

We thank the reviewers for their thoughtful comments and suggestions on our original manuscript. We have added new data and analyses to address each comment raised by the reviewers. These are incorporated throughout the revised manuscript (highlighted). Below, please find our point-by-point response (blue) to the comments. We believe that addressing these suggestions has significantly improved our manuscript.

Reviewers' Comments:

Reviewer #1:

Remarks to the Author:

Mattson et al. conducted a series of CRISPR screens to identify surface proteins essential for cancer cell survival using surface proteome library followed by integrin-focused library, and to explore potential drug target structure using tiling sgRNA library on ITGAV. They revealed a critical role of ITGAV-ITGB5 dimerization for cancer cell expansion and characterized an integral pocket within ITGAV's beta-propeller

domain for the dimerization. Further, they combined compound docking algorithm with tiling-sgRNA screens to identify a compound that inhibit the integrin signaling via heterodimer dissociation. In general, the high-throughput screens and computational methods used in this study are well designed and technologically sound. The method (CRISPR-TICA) that combines tiling screen and compound docking for drug discovery is interesting and hold potential for future applications. The manuscript is well-written with a clear storyline. Based on my expertise, the comments below are focused on the design, interpretation, and validation of high-throughput CRISPR screens, as well as the computational methods employed in this study.

Major:

1. The authors claimed that “ITGAV as a vulnerability in multiple cancers”. They made the statement based on i) ITGAV is essential for cancer cell viability; ii) high ITGAV expression is associated with poor survival prognosis. These lines of evidence, however, are insufficient to support the argument of “cancer vulnerability”. To confirm “cancer vulnerability”, the authors should demonstrate: a) ITGAV is over-expressed in at least a subset of patient tumor samples compared to normal cells; b) cancer cell dependency on ITGAV is correlated with its expression (I found this is true in Depmap data).

Thank you for the insightful suggestions. (a) We evaluated the GENT2 database which collected gene expression data from > 68,000 normal and cancer samples (<http://gent2.appex.kr/gent2/>). We found that ITGAV is over expressed in multiple cancer types including the ones tested in our cell surface proteome CRISPR screens in **Fig. 1A-C**, i.e., the breast, pancreas, brain, colon, and lung cancers (see below). We have now updated this new analysis and discussion in the **revised manuscript lines 122-123 and Suppl. Fig. S5**.

(b) We followed the reviewer’s recommendation and evaluated ITGAV’s cancer cell dependency in ~1,000 cancer cell models from the DepMap database (see to the right; each dot indicates a cell model). As indicated by the reviewer, the cancer cell dependency of ITGAV (x-axis; a more negative CERES score indicates a stronger survival dependency to ITGAV) is correlated with ITGAV’s expression level (y-axis). We have now updated this new analysis and discussion in the **revised manuscript lines 113-114 and Suppl. Fig. S2D (left panel)**.

2. The authors used a cell surface proteome library for the first screen. Since this is an unbiased large-scale screen, I'd expect more information to be provided and discussed on the screen, including: i) the QC outcome of fold-change by positive and negative controls, which is provided by MaGeCK (this can be shown in supplementary figure); ii) Other than ITGAV, what are the other top hits of surface proteome essential for cancer cells? Of note, ITGAV is essential for all cell lines. Are there other proteins essential for a subset of cell lines? These proteins may also be interesting to the readers. A summary of the screen hits is preferred such that the manuscript could provide an overview of the functional impact of surface proteins on cancer cell viability phenotype.

i) For the QC outcome of the surface proteome CRISPR screen: In addition to the fold-change of positive controls (yellow) and negative controls (green) showed in **Fig. 1C** (also see here, **panel A**). We now report the statistical analysis of the QC controls in **panel B**. These results indicate a significant separation of the positive (yellow) and negative (green) controls in the surface proteome CRISPR screens. We also include the complete data table in **Suppl. Table S3**.

ii) Further evaluation of the surface proteome CRISPR screens revealed that in addition to **ITGAV (hit #1)**, there are 2 other genes also essential to the cancer cells (**panel C**). **ATP6AP2 (hit #2)** is a v-type ATPase required for the proton transportation and acidification of the endo-lysosomal vesicles. **TFRC (hit #3)** is a transferrin receptor mediating the cellular uptake of iron. Both ATP6AP2 and TFRC are potential candidates for therapeutic targeting; however, compared to ITGAV, these additional candidates exhibit less correlation between their cancer cell dependency and their expression level in the DepMap analysis (**panel D**). We have now updated these new analyses in the **revised manuscript lines 113-114 and Suppl. Fig. S2A-D**.

Furthermore, we evaluated the top 10 candidate hits from each cell type and found that ITGAV and ATP6AP2 are the common essential surface proteins in these cell types (see below; red). Additionally, we observed genes that only hit a specific cell type (gene name color-coded). These hits represent candidate genes that could offer therapeutic opportunities in the select cancer types. We have now updated these new analyses in the revised manuscript lines 113-114 and **Suppl. Fig. S2E,F**.

Rank	MDA231	PANC1	U251	SW620	H661
1	TFRC	ITGAV	ITGAV	TFRC	ADGRL2
2	ATP6AP2	TFRC	ATP6AP2	ATP6AP2	HMMR
3	ITGAV	ATP6AP2	ITGB5	ULBP2	ITGB1
4	CD320	ITGB5	SLC4A7	CDH1	ERBB2
5	ART4	SLC7A5	LDLR	GGT1	ATP1B3
6	EVI2B	LDLR	TGFBR2	LDLR	CD164
7	ADGRL2	SLC12A9	TFRC	BSG	MFAP3
8	WNT5A	BMPR1A	EMB	ATRN	ATP6AP2
9	TRHR	KIAA1324L	PTPRJ	ITGB5	ITGAV
10	ITGB5	HMMR	LEPR	ITGAV	TRHR

3. Because ITGAV is the key protein in this study, a rescue experiment is necessary to validate its essentiality for cancer cell viability, in addition to the KO experiments shown in Figure 1.

We agree with the reviewer’s comment that a rescue experiment could strengthen our study. For this, we first evaluated the sgITGAV targeted position and found that the recognition sites of sgITGAV#2 and sgITGAV#3 span across the exon-intron junctions (see below, **panel A**). These sgITGAVs only target the endogenous ITGAV but cannot recognize the ITGAV cDNA sequence (w/o introns), thus allowing the reconstitution of ITGAV through cDNA transduction. Our results revealed that the transduction of exogenous ITGAV cDNA in MDA231 cells significantly reversed the impact of sgITGAV on cell survival (**panel B**), validating the roles of ITGAV in cancer cell maintenance. We have now updated these new results in the revised manuscript lines 119-120 and **Suppl. Fig. S4**.

4. To me, the CRISPR-TICA method is a simple combination of tiling-sgRNA screen and docking algorithm, which is not completely novel in methodology. Nevertheless, the study provides a good prototype of utilizing tiling-sgRNA screen for drug discovery. To demonstrate its potential utility for future applications, authors could provide a few more examples in which a tiling screen highlight the known drug-targeting pockets. This can be done by performing similar analysis on published tiling-sgRNA screen data (e.g. PMID 27260157). Such examples will strengthen the impact and generality of the method.

We would like to thank the reviewer for this insightful suggestion! Following this idea, we obtained the CRISPR tiling data from Munoz et al. (PMID: 27260157) and identified 4 proteins with well-defined drug-targeting pockets. These include the bromodomain (a chromatin binding module) of BRD4 and the kinase catalytic cores of AURKB, CDK1, and WEE1 (see below). The green boxes highlight the CRISPR-TICA region of interest based on the CRISPR sensitivity. The yellow arrows indicate the previously reported inhibitors for these 4 proteins. These analyses revealed the utility of CRISPR-TICA as a generalizable approach for drug discovery. We have now updated these new results and discussions in the revised manuscript lines 324-328 and Suppl. Fig. S14.

5. For the mutagenesis analysis with NanoBRET (Fig. 4F), negative controls with mutations of aromatic residues outside the pocket are preferred to demonstrate the critical function of the pocket for dimerization.

This is an excellent suggestion. For this, we located 4 other aromatic residues (see below; W234, F507, W790, and F938) across the extracellular regions of ITGAV. Compared to the β -propeller HIP residues presented in **Fig. 4F**, alanine substitution of the aromatic residues outside of HIP did not impair the ITGAV/ITGB5 dimerization measured by the NanoBRET assay. We have now updated these important negative control data in the **revised manuscript lines 198-199 and Suppl. Fig. S9**.

6. Ideally, it is preferred to show the crystal structure of ITGAV-ITGB5 interaction via the identified pocket. If this is too laborious for the revision, an alternative way is to use AlphaFold2 to predict the interaction interface.

Thanks to the reviewer's suggestion. Here, we use the AlphaFold2 to model the extracellular domain of ITGB5 (**panel A**, cyan) and overlay with the ITGB3 portion of integrin $\alpha\beta3$ structure resolved by Xiong et al. (PDB ID: 3IJE, chain B; **panel A**, blue). Overall, we observed high concordance of the 3D structures between the ITGB3 and ITGB5, including the highly conserved basic amino acid (ITGB3's R287 or ITGB5's K287) in the loop motif of the β A domain (**panel B**).

We further used the predicted ITGB5 structure (cyan) to model its interaction with the ITGAV portion of integrin $\alpha\beta3$ structure resolved by Xiong et al. (PDB ID: 3IJE, chain A; red; **panel C**). Molecular dynamics simulations of the ITGAV's β -propeller (**panel D**; red) and ITGB5's β A-domain (cyan) indicated a close contact between ITGB5's K287 (green) and multiple aromatic residues on ITGAV's HIP pocket (purple box), forming cation- π interactions. Importantly, the occupancy of Cpd_AV2 (**panel E**; yellow) into ITGAV's HIP pocket disengaged the side chain of ITGB5's K287 from stably interacting with ITGAV's HIP pocket. To confirm the prediction from the molecular dynamics simulations, we mutated K287 to an alanine. We found K287A significantly attenuated the ITGAV/ITGB5 NanoBRET signal (**panel F**), highlighting an essential role of the ITGB5's K287 in integrin $\alpha\beta5$ assembly. We have now updated these new results in the revised manuscript lines 240-247 and Suppl. Fig. S12.

Minor:

1. It seems to me that the 2nd screen is sort of redundant to the 1st screen because the genes and cell lines were included in the 1st screen. What's the rationale of performing the 2nd screen? Is there any hit identified in the 2nd screen but not in the 1st screen due to the difference of guide per gene?

In the 1st screen (surface proteome; ~580 genes with 5 sgRNAs/gene), we identified ITGAV as the top essential surface gene. However, which β integrin serve as the critical partner of ITGAV was not clearly revealed. We, therefore, performed the 2nd screen focusing only on the 26 integrin family genes with 25 sgRNAs/gene. The higher sgRNA number per gene increased the statistical confidence (as compared to the 1st screen) and allowed the identification of ITGB5 as the top essential β integrin (**panel A**; y-axis). Similar to ITGAV (see revised Suppl. Fig. S2D, left panel),

we also observed that the cancer cell dependency on ITGB5 is correlated with its expression (**panel B**; source: DepMap database), highlighting the role of integrin $\alpha\text{V}\beta\text{5}$ in cancer cell maintenance. Notably, while the 1st screen didn't observe ITGB5 within the leading edge essential genes, re-evaluating of this surface proteome screen revealed that ITGB5 remains the 2nd most essential integrin (**Panel A**; x-axis). We have now updated these new analyses and discussions in the revised manuscript lines 155-156 and Suppl. Fig. S7.

2. Font format and size are inconsistent. Some are too big, and some are too small. Reformatting is necessary before publication.

Thank you for bringing this to our attention. We have now revised the font format/size of the figures to improve the legibility. We will continue to work with the editors and follow the journal guidelines during the final production.

Reviewer #2:

Remarks to the Author:

A. Summary of key results

Mattson et al. use an unbiased CRISPR KO screening technology and identify Integrin alphaVbeta5 ("avb5") as a cell surface protein preventing cancer cell line apoptosis. KO of RAC1 (a known avb5 downstream partner) had a similar effects. Elegant CRISPR gene-tiling identified hydrophobic residues surrounding the central pore of the ITGAV chain β -propellor as necessary for avb5 activity. Alanine-substitution of these residues prevented alphaV and beta5 chains from forming heterodimers. Molecular docking of an 1E5 compound library onto this region identified model leads, and these reduced cell viability of MDA-MB-231 cells. One compound, AV2, clearly reduced cell surface expression of alphaVbeta5, and at similar concentration induced apoptosis and cell rounding.

Recombinant alphaV interacted with Cpd-AV2, and AV2 reduced cell surface expression of avb5 and cell viability compared to the avb5/avb3 competitive inhibitor cilengitide. The authors conclude that their method identifies novel integrin inhibitors based on an ability to disrupt alpha-beta chain dimerization.

B. Originality and significance

That avb5 can protect tumor-derived cells from apoptosis is known (e.g. Uhm et al. Clin Cancer Res (1999) 5 (6):1587; Lane et al. Oncogene (2010)29:3519), but the CRISPR unbiased screen confirming this is novel. The original CRISPR mediated search for novel inhibitors of integrin function finding regions on the alpha-v chain critical for dimerization is a novel and promising approach. And the molecular docking screen identifying candidates which trigger apoptosis via these regions sites is elegant and original. Overall, the team attempt to show that this approach in a general unbiased way can identify an so far unexpected targetable region on integrins, and use docking to credibly identify small molecules that can bind this region.

Thank you for these insightful comments. We have now included the references indicating the roles of integrin α V β 5 in protecting the cancer cells from apoptosis (PMID: 10389948, PMID: 20400979) in the revised manuscript lines 261-273.

C. Data and methodology

The data is well described and generally well presented. The methodology is certainly original, and the authors show very well once again the power of CRISPR screening approaches. Adequate positive and negative controls are included in the screen. It is the unbiased nature of their screens that make them interesting. Although the target maybe less so, recalling that at least one drug, the pan-av inhibitor antibody Abituzumab had minimal observable avb5-mediated effect (Elez et al. Ann Oncol. (2015);26(1):132).

The approach seems valid, but perhaps the interpretation of the effect less so.

Thank you for these evaluations. We have now included the reference of pan- α V inhibitory antibody Abituzumab (PMID: 25319061) in the revised discussions at line 289.

D, Appropriate use of statistics and treatment of uncertainties.

No issues.

E. Conclusions: robustness, validity, reliability

[E.1] May we start with integrin dimerization? The authors claim that by messing up the hydrophobic rings of the β -propellor they affect dimerization. They have an excellent carefully constructed dimerization FRET-like assay, NanoBRET easily able to detect dimerization, and they show that alanine substituting their carefully discovered CRISPR identified hydrophobic residues (Fig 4F) disrupt dimerization. The cage of hydrophobic β -propellor residues enclosing the Arg261 (avb3)/Lys(b1,b5,b6,b8) residue of the beta-chains and likely holding them in place was noted in our original structure paper. This is a direct demonstration that its disruption affects heterodimerization. But they do not show that their apoptosis inducing compounds especially Cpd-AV2 disrupts dimerization. Alanine scan- disrupting the selected hydrophobic residues may prevent en-caging of the necessary beta-chain Lys261 residue (the postulated inhibition of dimerization). Or the alphaV chains aren't folding correctly, and no β -propellor exists. It may well be that Cpd-AV2 indeed disrupts dimerization. But it is not shown.

As highlighted by the reviewer, based on the original $\alpha\beta 3$ structural paper (Xiong et al, 2001 Science), the cage (i.e., our HIP pocket) of hydrophobic β -propeller residues enclosing the Arg261/Lys261 residue (R287/K287 in our alignment) of the integrin β -chains is likely required for stabilizing the integrin α/β heterodimers. To examine this hypothesis and evaluate the $\alpha\beta 5$ interaction, we utilized the AlphaFold2 algorithm to model the extracellular domain of ITGB5 (**panel A**, cyan) and overlay with the ITGB3 portion of integrin $\alpha\beta 3$ structure resolved by Xiong et al. (PDB ID: 3IJE, chain B; **panel A**, blue). Overall, we observed high concordance of the 3D structures between the ITGB3 and ITGB5, including the highly conserved basic amino acid (ITGB3's R287 or ITGB5's K287) in the loop motif of the βA domain (**panel B**).

We further used the predicted ITGB5 structure (cyan) to model its interaction with the ITGAV portion of integrin $\alpha\beta 3$ structure resolved by Xiong et al. (PDB ID: 3IJE, chain A; red; **panel C**). Molecular dynamics simulations of the ITGAV's β -propeller (**panel D**; red) and ITGB5's βA -domain (cyan) indicated a close contact between ITGB5's K287 (green) and multiple aromatic residues on ITGAV's HIP pocket (purple box), forming cation- π interactions. Importantly, the occupancy of Cpd_AV2 (**panel E**; yellow) into ITGAV's HIP pocket disengaged the side chain of ITGB5's K287 from stably interacting with ITGAV's HIP pocket. To confirm the prediction from the molecular dynamics simulations, we mutated K287 to an alanine. We found K287A significantly attenuated the ITGAV/ITGB5 NanoBRET signal (**panel F**), highlighting an essential role of the ITGB5's K287 in integrin $\alpha\beta 5$ assembly. We have now updated these new results in the revised manuscript lines 240-247 and Suppl. Fig. S12.

Although the artifacts such as disruption of protein folding could mislead the conclusion of the mutagenesis experiments, we think the reciprocal alanine substitutions (i.e., mutations in either ITGAV's cage residues or ITGB5's K287) and their NanoBRET results increased the confidence of our proposed model. We now also examined the capacity of Cpd_AV2 in disrupting the $\alpha\beta 5$ interaction by NanoBRET in our response to the reviewer's suggestion [F.1] (see section F.1 below).

[E.2] The use of an EColi derived alpha chain is odd. Integrin avb5 is commercially available. As far as I am aware, intact integrin chains have not been shown to fold correctly in bacteria. Given the high degree of glycosylation this is no big surprise. Only mammalian recombinant integrins have demonstrably correct folding and activity. Does the authors' "refolded" chain interact with conformation-dependent anti-av antibodies, for example? The target compounds, like AV2, look pretty hydrophobic, so perhaps it is not surprising that they interact (Fig6A-C). with hydrophobic patches on the refolded E.coli recombinant alpha chain. In short I have doubts about the validity of the conclusion that the selected reagents affect dimerization.

We agree with the reviewer's insightful comment! Indeed, we initially used the commercially available full-length human ITGAV/ITGB5 heterodimer protein (**panel A**; ordered from Acro Biosystems) for the thermal shift assay. We observed that Cpd_AV2 (40 μ M) slightly increased the melting temperature (T_m) of the ITGAV/ITGB5 heterodimer by 1.9 $^{\circ}$ C (**panel B**). Interestingly, the Cpd_AV2 treated melting curve revealed a 2nd melting peak, indicating a potentially multiphase thermo-denaturation of the full-length ITGAV/ITGB5 heterodimer. This might be due to the involvement of multiple globular domains in the full-length ITGAV/ITGB5 heterodimer (combined molecular weight \sim 200kDa). After consulting with our colleagues in structural biology and drug development, we were convinced that the full-length ITGAV/ITGB5 heterodimer is likely too big for the thermal shift assay. The multi-domains included in the full-length ITGAV/ITGB5 also raised the challenges of determining where inside the full-length ITGAV/ITGB5 the Cpd_AV2 is binding.

To examine the binding of Cpd_AV2 specifically to ITGAV's β -propeller domain, we were recommended to purify the β -propeller domain from E. coli using the strategy reported in Ruthenburg et al. 2006 Nat. Struct. Mol. Biol. (PMID: 16829959; expression of WDR5's β -propeller domain). In our experiment, β -propeller domain of ITGAV (F31 – S492; \sim 50kDa) plus a His-tag was expressed from E. coli for purification (**Fig. 6A**). This approach allowed us to selectively evaluate the potential of Cpd_AV2 binding on the purified ITGAV's β -propeller domain (**Fig. 6B**).

We agree with the reviewer that this biochemical approach might be affected by the generally non-glycosylated nature of the bacterially produced proteins. On the other hand, we think the combination of cell surface α V β 5 flow cytometry (detecting the endogenous full-length ITGAV/ITGB5 heterodimer) and NanoBRET (measuring the exogenous full-length ITGAV/ITGB5 interaction) utilized in our manuscript enabled us to evaluate the impact of Cpd_AV2 on the full-

length glycosylated integrin $\alpha V\beta 5$ expressed on the human cells. We have now updated these discussions in the revised manuscript lines 307-312.

[E.3] Integrin avb5 as critical: the cell culture conditions used in the selection assays ensure that a vitronectin substrate is offered to the cells, which all predominantly (4 epithelial derived; one avb5/avb3 glioma) use avb5 to attach to vitronectin. So removing this integrin by KO forces cells to round up, as shown, and sometimes to detach. This would tend to favour apoptosis.

We have now included these discussions in the revised manuscript lines 271-273.

F. Suggested improvements, experiments data for possible revision.

[F.1] The authors should please show the readers the NanoBRET assay in at least MDA-MB-231 cells exposed to various concentrations of Cpd-AV2 with a time course. This would directly show whether AV2 induces loss of dimerization.

We agree with the reviewer's comment that examining the effect of Cpd_AV2 on the NanoBRET assay is necessary. As multiple incubation steps are involved in NanoBRET assay (e.g., incubations with HaloTag ligand, luciferase substrate, etc.), this method is more suitable for measuring the steady state ITGAV/ITGB5 heterodimer interaction (e.g., between different ITGAV or ITGB5 variants). To observe the dynamic changes of ITGAV/ITGB5 heterodimer level upon Cpd_AV2 treatment, we first utilized the $\alpha V\beta 5$ flow cytometry to monitor the amount of ITGAV/ITGB5 heterodimer on the cell surface from 0.5 min to 64 min after 40 μM Cpd_AV2 treatment (**panel C**). These results indicated that a 1-hour Cpd_AV2 incubation is sufficient to disrupt the ITGAV/ITGB5 heterodimers.

Based on this condition, we examined the effect of Cpd_AV2 on NanoBRET (**panel D**), validating the capacity of Cpd_AV2 to destabilize integrin $\alpha V\beta 5$ heterodimer. We have now updated these new results in the revised manuscript lines 225-226 and Suppl. Fig. S11C,D.

[F.2] Antibodies. Please could the authors refer to protein concentrations of the antibodies they use, not dilutions? The readers have no idea of the starting concentrations, so the dilutions are sadly meaningless.

Thank you for this suggestion. We have now updated the antibody concentrations in the revised manuscript lines 452-453 and Suppl. Fig. S16.

Antibody Name	Catalog ID	Stock Concentration	Application (dilution)	Working Concentration
Rabbit anti-ITGAV	4711, CST	53 µg/ml	Western blot (1:1,000)	53 ng/ml
Rabbit anti-ITGB5	3629, CST	338 µg/ml	Western blot (1:1,000)	338 ng/ml
Rabbit anti-RAC1	4651, CST	129 µg/ml	Western blot (1:1,000)	129 ng/ml
Mouse anti-β-actin	ab8226, Abcam	1 mg/ml	Western blot (1:5,000)	200 ng/ml
Goat anti-mouse (HRP-conjugated)	31430, Invitrogen	0.8 mg/ml	Western blot (1:10,000)	80 ng/ml
Goat anti-rabbit (HRP-conjugated)	31460, Invitrogen	0.8 mg/ml	Western blot (1:10,000)	80 ng/ml
Mouse anti-integrin αVβ5 (clone P1F76)	sc-13588, Santa Cruz	200 µg/ml	Flow cytometry (1:200)	1 µg/ml
Donkey anti-mouse (AF488-conjugated)	ab150119, Abcam	2 mg/ml	Flow cytometry (1:500)	4 µg/ml
Mouse anti-integrin αVβ5 (clone P1F6; AF647-conjugated)	920005, Biolegend	100 µg/ml	Flow cytometry (1:100)	1 µg/ml

[F.3] I was puzzled that the authors used an obscure antibody for the critical cell-surface FACS data investigating avb5 expression following compound treatment. I would suggest a well characterized antibody (perhaps P1F6?) in this crucial assay would be more definitive?

We initially selected the clone P1F76 antibody (sc-13588) for integrin αVβ5 detection based on the data reported in 2015 Cardiovasc Res (PMID: 25952901), 2018 EBioMedicine (PMID: 29907328), etc. We also performed an in-house QC experiment to confirm the sensitivity of clone P1F76 antibody to the genetic ablation of ITGAV (shown in Fig. 5E).

Here, we would like to thank the reviewer's suggestion to use another well-characterized integrin αVβ5 antibody (clone P1F6) to cross-validate the anti-αVβ5 (clone P1F76) flow cytometry results. We found that Cpd_AV2 could block the cell surface integrin αVβ5 signal with similar IC₅₀ values (6.9 µM by P1F76 and 6.1 µM by P1F6), increasing the confidence of the data reported in Fig. 5F. We have now updated these new results in the revised manuscript lines 225-226 and Suppl. Fig. S11A,B.

[F.4] The Western blots showing gel slices should please be complemented by images of the entire gel blots, so that the readers may assess the specificity of the antibodies used directly.

We thank to reviewer's reminder to present the entire gel blots to allow evaluation of the antibody specificity. We have now updated these blots in the revised manuscript line 452-453 and Suppl. Fig. S17.

[F.5] In Fig2H and 5J the authors mention "Cell dimension". I think they may mean projected cell area/diameter. Please could they clarify?

We would like to thank reviewer for this excellent note. Our initial evaluation of the cell dimension focused on measuring the average length (μm) of each cell from three different angles. We think the reviewer's recommendation of measuring the projected cell area (i.e., cell size measured by ImageJ; μm^2) could better represent the morphological changes of cells upon ITGAV inhibition. We have now updated these new analyses in the revised Fig. 2H and 5J.

[F.6] According to the authors hypothesis, the Cpd-AV5 interacts with the alphaV chain. So it is expected that it would disrupt heterodimers of at least avb1, avb6 and avb8 too, they also use Lys encapsulated in

the alphaV chain b-propeller. Might a proof of concept involve disrupting an avb6 dependent adhesion (HT29 cells on fibronectin springs to mind)?

We thank the reviewer's suggestion to examine the impact of our study on additional ITGAV integrins. As pointed out by the reviewer, the basic amino acid encapsulated in ITGAV's beta-propeller is conserved across the ITGB1/3/5/6/8 genes (**panel A**; K/R287), highlighting the potential of Cpd_AV2 disrupt the functions of alphaVbeta1, alphaVbeta3, alphaVbeta5, alphaVbeta6, and alphaVbeta8 integrins. As a proof of concept, we monitored the integrin alphaVbeta6-dependent cell adhesion of HT-29 cells (a colorectal carcinoma) to fibronectin (Fn) reported by Kemperman et al. (PMID: 9223381). We found that pre-incubation of the HT-29 cells with Cpd_AV2 led to a dose-dependent blockade of HT-29 cell adhesion to the Fn-coated wells (**panel B**). We have now updated these new results and discussions in the revised manuscript lines 300-305 and **Suppl. Fig. S13**.

G. References.

Appear appropriate

H. Clarity and context.

The work is well written and described, with the exceptions noted.

Decision Letter, first revision:

Message: Our ref: NSMB-A47687A

12th Oct 2023

Dear Dr. Chen,

Thank you for submitting your revised manuscript "CRISPR-tiling instructed structural genetic screen identifies a novel class inhibitor that disrupts the integrin heterodimer stability" (NSMB-A47687A). It has now been seen by the original referees and their comments are below. The reviewers find that the paper has further improved in revision, and therefore we'll be happy to accept it in principle in Nature Structural & Molecular Biology, pending minor revisions to comply with our editorial and formatting guidelines.

We are now performing detailed checks on your paper and will send you a checklist detailing our editorial and formatting requirements in about 2 weeks. Please do not upload the final materials and make any revisions until you receive this additional information from us.

Sincerely,

Dimitris Typas
Associate Editor
Nature Structural & Molecular Biology
ORCID: 0000-0002-8737-1319

Reviewer #1 (Remarks to the Author):

This is a revised version of the manuscript. I appreciate the authors' time and efforts to address my concerns raised in the first round of review. All my concerns and questions have been clearly addressed and I'm satisfied with the revision.

Reviewer #2 (Remarks to the Author):

The authors have worked very hard to answer my initial comments, and especially the answer to query F. concerning NanoBret demonstration of dimerization - and the use of a well known anti-avb5 antibody PIF6 are enlightening.

I am reasonably convinced that their screen identify dimerization-sensitive sites and the resulting compound can inhibit it.

I think that the paper is now considerably strengthened and is a good study appropriate for publication.

Simon Goodman

Author Rebuttal, first revision:**Reviewer #1:**

Remarks to the Author:

This is a revised version of the manuscript. I appreciate the authors' time and efforts to address my concerns raised in the first round of review. All my concerns and questions have been clearly addressed and I'm satisfied with the revision.

We highly appreciate the reviewer's critiques and comments. We think that addressing these suggestions has significantly improved our manuscript.

Reviewer #2:

Remarks to the Author:

The authors have worked very hard to answer my initial comments, and especially the answer to query F. concerning NanoBret demonstration of dimerization - and the use of a well known anti-avb5 antibody PIF6 are enlightening.

I am reasonably convinced that their screen identify dimerization-sensitive sites and the resulting compound can inhibit it.

I think that the paper is now considerably strengthened and is a good study appropriate for publication.

Simon Goodman

We highly appreciate the reviewer's critiques and comments. We think that addressing these suggestions has significantly improved our manuscript.

Final Decision Letter:

Message 2nd Jan 2024

:
Dear Dr. Chen,

We are now happy to accept your revised paper "A novel class of inhibitors that disrupts the stability of integrin heterodimers identified by CRISPR-tiling instructed genetic screens"

for publication as an Article in Nature Structural & Molecular Biology. We sincerely apologise once more for the delay in processing your manuscript during the final steps of the process.

Your paper will be published online soon after we receive proof corrections and will appear in print in the next available issue. You can find out your date of online publication by contacting the production team shortly after sending your proof corrections.

You may wish to make your media relations office aware of your accepted publication, in case they consider it appropriate to organize some internal or external publicity. Once your paper has been scheduled you will receive an email confirming the publication details. This is normally 3-4 working days in advance of publication. If you need additional notice of the date and time of publication, please let the production team know when you receive the proof of your article to ensure there is sufficient time to coordinate. Further information on

our embargo policies can be found here:
<https://www.nature.com/authors/policies/embargo.html>

Please note that *Nature Structural & Molecular Biology* is a Transformative Journal (TJ). Authors may publish their research with us through the traditional subscription access route or make their paper immediately open access through payment of an article-processing charge (APC). Authors will not be required to make a final decision about access to their article until it has been accepted. Find out more about Transformative Journals <https://www.springernature.com/gp/open-research/transformative-journals>

You will not receive your proofs until the publishing agreement has been received through

our system.

Sincerely,

Dimitris Typas
Associate Editor
Nature Structural & Molecular Biology
ORCID: 0000-0002-8737-1319